# The rewiring of a terminal selector regulatory cascade generates convergent neuronal laterality

**Dylan L. Castro**, **Ivan M. Dimov**, **Marisa Mackie**, **Heather R. Carstensen**, **Mary T. Barsegyan**, **Ray L. Hong***

Department of Biology, California State University Northridge, Northridge, California, Unites States of America

* ray.hong@csun.edu

## Abstract

Neuronal identity is established and maintained by "terminal-selector" transcription factors, yet how these networks evolve remains unclear. We examined the specification of the chemosensory ASE and thermosensory AFD neurons in the nematode *Pristionchus pacificus*, a species that expresses the terminal-selector, *Ppa*-CHE-1, in both sensory neurons. To determine if the ASE neurons exhibit left-right laterality, we used HCR-FISH and transgenic reporters to discover 8 ASE left-right-specific and 3 AFD-specific receptor-type guanylyl-cyclases. Late embryos exhibit a multipotential state in which AFD precursors transiently co-express all three types of ASEL, ASER and AFD markers. A forward genetic screen for defects in ASER asymmetry identified a *Ppa*-DIE-1 homolog, whereas targeted mutations revealed the maintenance of AFD neuronal identity requires another terminal-selector, *Ppa*-TTX-1, and CNG channels, *Ppa*-TAX-2/TAX-4. Mutations in the microRNA *miR-8345* and *pash-1* responsible for miRNA-processing convert ASEL to ASER fate while changes to other conserved regions in the 3′ UTR of the *cog-1* homolog reveal multiple sites that act as a toggle between left/right ASE versus AFD identities. Together, these results demonstrate that *P. pacificus* deploys a miRNA-mediated regulatory repertoire to generate three distinct neuronal fates through the *Ppa-cog-1* 3′ UTR as a key regulatory nexus.

## Author summary

Transcription factors known as terminal selectors help neurons acquire and maintain their specific identities, such as sensing soluble chemicals and temperature. In nematodes, while most sensory neurons develop together in pairs, the chemosensory neurons further exhibit left-right laterality in chemosensory receptor expression. Surprisingly, we discovered that even the thermosensory neurons not destined for chemosensory receptor expression show potential for multiple terminal fates, transiently co-expressing chemosensory markers early in

**Data availability statement:** The dataset and tabulation script is available on GitHub. https://github.com/honglabcsun/Neuronal_Asymmetry.

**Funding:** This study was supported by the National Institutes of Health SC1GM140970 to RLH. The funders had no role in study design, data collection and analysis, decision to publish, or preparation of the manuscript.

**Competing interests:** The authors have declared that no competing interests exist.

embryogenesis before restricting their expression to only thermosensory receptors. Like the nematode *C. elegans*, *P. pacificus* uses microRNA-based regulation to produce distinct left-right neuron sub-types. Unlike *C. elegans* however, *P. pacificus* employs multiple post-transcriptional regulators in a conserved negative feedback loop as a toggle switch between 3 distinct terminally differentiated developmental fates. Since little is known about how miRNA and their targets evolve, this study highlights a recurring miRNA regulatory architecture that may improve our understanding of nervous system evolution.

## Introduction

Neurons in humans and other animals acquire their individual features during embryogenesis and continue to differentiate post-mitotically. During the last phase of differentiation, neuron type-specific gene batteries specify terminal identity genes (also known as "effector genes") to confer the stable functional properties that distinguish each neuron type such as neurotransmitter biosynthesis, neuropeptides, neuronal connectivity, ion channels, and membrane receptors. The coordinated expression of these terminal identity genes is controlled by the actions of terminal selectors, which are transcription factors (TFs) necessary for the establishment and maintenance of neuronal identity [1,2]. While the evolutionarily conserved functions of terminal selectors have been found across taxa, including in *Drosophila* and mouse neurons, the best-studied genetic system for their role in neuronal patterning at the single neuron level has been the nematode *Caenorhabditis elegans* [3]. Terminal selectors, mostly homeobox genes in *C. elegans*, have been identified for 111 of the 118 neuron types [3,4]. Yet, the opportunity to leverage this *C. elegans* paradigm to examine how terminal selectors accommodate changes in neuron type specification in other nematodes has not been explored.

The emerging model system *Pristionchus pacificus* and *C. elegans* share nearly 1–1 neuronal homology but are different in connectivity [5–7], which provides a well-defined comparative system to interrogate how homologous genes interact within neuronal constraint to generate behavioral diversity. While both species eat bacteria, *P. pacificus* is an omnivorous and predatory nematode species that is capable of killing larvae of other nematode species, as well as *P. pacificus* conspecifics [8–10]. *P. pacificus* and *C. elegans* also have divergent chemosensory profiles in odor preferences and in taste recognition, possibly due to the unique natural ecology of *P. pacificus* with insects and insect-associated bacteria that differ from the compost-associated *C. elegans* [11–16]. Despite nearly identical neuronal types between the two species, the molecular mechanisms underlying the divergent behaviors of *P. pacificus* remain poorly understood.

When the expression pattern of terminal selectors in homologous neurons evolve, how do downstream regulators adjust their roles to maintain the target effector gene profiles? Some terminal selectors, such as homeobox TFs, achieve specificity by combinatorial interactions with their co-factors, while other terminal selectors, such as

the zinc-finger TF CHE-1 (homolog of Drosophila GLASS), achieve specificity by limiting their expression only to the neurons where they act in their terminal differentiation role. One of the key functions of *che-1* is to specify the ASE chemosensory neurons, one of 12 amphid sensory neuron pairs in the head ganglia of *C. elegans* [17–19]. What sets CHE-1 apart, however, is its function to further promote lateral asymmetry in the ASE neurons, which results in the left and right ASE neurons that differentially express receptor-type guanylyl cyclases (rGCs, encoded by *gcy* genes) [20,21]. This asymmetry enables the two ASE neurons to detect distinct sets of salt ions [22–24]. Thus, CHE-1 is a part of a gene regulatory network that specifies both ASE as well as the ASE-subtype lateralized identities, with the microRNA *lsy-6* occupying a pivotal upstream position in this network [25–28].

Despite this well-characterized mechanism in *C. elegans*, the evolutionary origins and generality of ASE laterality remain unclear. The short size of miRNA (~22-nt), lineage-specific evolution, frequent de novo emergence, and complex target recognition mechanisms conspire to make homology assignment of miRNAs particularly challenging. Notably, sequence outside of the seed region of the *lsy-6* miRNA is poorly conserved in mature miRNAs in non-*Caenorhabditis* nematode species such as in the *Strongyloides ratti* and *Pristionchus pacificus* [29]. Furthermore, *P. pacificus* does not exhibit the same expansion of ASEL- and ASER-specific *gcy* genes observed in *C. elegans* [6]. Previously, functional comparisons of miRNA and ASE-specific *gcy* genes were confined to within *Caenorhabditis* species [21,25,30], while the divergence time between *C. elegans* and *P. pacificus* is estimated to be much deeper ~300 million years [31]. Based on these genomic differences, it was previously hypothesized that ASE neurons in *P. pacificus* lack lateralized identity [6]. However, our recent findings prompt a re-evaluation of this view. Calcium imaging of neuronal activity in *P. pacificus* demonstrated that this species exhibits distinct, lateralized ASE responses, mediated in part by an ASER-specific *Ppa-gcy-22* paralog, *Ppa-gcy-22.3* [16]. More intriguingly, the co-expression of homologs of the two terminal selectors for ASE and AFD identity, *che-1* and *ttx-1,* in the *P. pacificus* AFD neurons provides an example of the specification of AFD neurons co-opting the ASE regulatory network. These findings suggest an exaptation of the CHE-1-mediated regulatory network from ASE lateral fate specification to AFD fate determination. Consistent with this scenario, miRNA copy number and chromosomal arrangement have diverged significantly in *P. pacificus* compared to *C. elegans* [32,33]. Understanding how miRNAs insinuate themselves into an evolving regulatory architecture will deepen our understanding of convergent gene regulatory networks.

If miRNA regulation establishes ASE laterality in *P. pacificus*, are homologs of its regulators and targets conserved? In *C. elegans, cog-1* is an ortholog of the human Nkx6 homeobox gene that acts in the ASER to repress ASEL fate. COG-1 acts together with the UNC-37/Groucho co-repressor to repress *die-1*, which encodes for a zinc-finger transcription factor [19,34]. Likewise, in the ASEL, *cog-1* is negatively regulated by the miRNA *lsy-6,* which then indirectly allows for the expression of *die-1* to repress ASER fate in the ASEL. This double negative interaction produces a mutually exclusive expression of either *cog-1* or *die-1* in the ASEL and ASER [19,25,27,35]. Mutations in *lsy-6, cog-1,* or *die-1* result in the transformation of one asymmetric ASE subtype to another, such that there could be "2xASEL" or "2xASER" symmetric phenotypes, as well as partially asymmetric states with hybrid ASE neurons in *fozi-1* or *lim-6* mutants [28,36,37]. The negative regulation of *cog-1* by *lsy-6* miRNA is mediated by the *cog-1* 3' UTR, which contains two *lsy-6* binding sites [25,36]. It has been proposed that one reason for the presence of both transcriptional and miRNA-mediated post-transcriptional control in *Cel-cog-1* is because the maturation of *C. elegans* ASEs is very sensitive to the correct dosage of *cog-1* [19], hence the identification of a homologous network offers a valuable comparison.

If the miRNAs and their cognate binding sites are labile in regulatory networks underpinned by homologous regulators such as *die-1* and *cog-1*, does miRNA regulation change relative to transcriptional regulation, or do lineage-specific miRNAs emerge to re-establish their governance? With the expansion of *Ppa-che-1* expression, how does its regulatory cascade include the specification of both the asymmetric ASE neurons as well as the symmetric AFD neurons? These observations raise fundamental questions about the developmental logic and evolutionary plasticity of lateralized sensory systems in nematodes and underscore the existence of alternative, as yet unidentified, molecular strategies for

establishing neuronal asymmetry outside of *C. elegans*. We systematically identified the terminal identity genes in *P. pacificus* exclusively expressed in the AFD and ASE neurons by examining *gcy* genes using recent advances in *in situ* hybridization technology. This resulted in the adoption of a trio of *gcy* marker genes for the AFD, ASEL, and ASER neurons. Next, in a forward genetic screen using the asymmetrically-expressed *Ppa-gcy-22.3* reporter, we isolated a key regulator for ASEL identity– a homolog of *die-1*– that acts as a genetic switch between AFD and ASE identity by repressing ASE fate initially present in the late embryonic stage. Finally, we show evidence for miRNA regulation by using mutants that have defects in miRNA biogenesis and in putative miRNA binding sites in the *cog-1* homolog. Our work traces the outlines of a familiar cast of regulatory factors interacting in unfamiliar ways.

## Results

### CHE-1-dependent *gcy-7* and *gcy-22* paralogs mark ASEL and ASER neurons

The *P. pacificus* genome lacks discernible homologous terminal identity genes expressed by the *C. elegans* ASE neurons, such as the ASEL-type (*gcy-6, gcy-14, gcy-17, gcy-20*) and ASER-type (*gcy-1, gcy-2, gcy-3, gcy-4*) receptor-type guanylyl cyclases (rGCs) [20,21]. The lack of left-right receptor-type effectors lead to the initial interpretation that although the canonical CHE-1-binding "ASE" sites have been found in putative regulatory regions of *P. pacificus* rGCs, lateral asymmetrical expression of rGCs in the ASE neuronal homologs does not occur in *P. pacificus* [6]. However, upon closer inspection, we found 1-to-many homologs of *gcy-7* (ASEL-type) and *gcy-22* (ASER-type) are present in the *P. pacificus* genome. Phylogenetic analysis using the longest amino acid sequences of 30 of the 35 rGCs in the *P. pacificus* genome show clear clustering of the *P. pacificus*-specific *gcy-7* and *gcy-22* subfamilies (Fig 1). Thus, duplication events among the three *gcy-7* and five *gcy-22* paralogs in *P. pacificus* likely arose repeatedly as well as independently after the separation of *Pristionchus* and *Caenorhabditis* species. While all three *P. pacificus gcy-7* paralogs are on Chromosome IV, the five *gcy-22* paralogs are dispersed across 4 loci over 3 chromosomes (Chr. I, IV, X). Most interestingly, while *gcy-22.1* and *gcy-22.2* on Chromosome I are likely to be the result of recent duplications, the next closest paralog, *gcy-22.3*, is located on Chromosome IV and is the best-reciprocal BLAST hit of the *C. elegans gcy*-22. Since the *P. pacificus* Chromosome I is likely the result of an ancient translocation between the ancestral *C. elegans* Chromosome V and X [38], the two *gcy-22* paralogs (*gcy-22.3* and *gcy-22.4)* on Chromosome IV could represent additional intrachromosomal rearrangements during the evolution of *Pristionchus* species. Lastly, there are two *P. pacificus gcy-5* paralogs, but *Ppa-gcy-5(PPA13334)* is more likely to share ancestry with the *Caenorhabditis gcy-5* ortholog given their synteny on Chromosome II, compared to PPA02210 that is located on Chromosome IV. Based on the clustering of these 9 rGCs in the gene phylogeny between the *Caenorhabditis gcy-7* and *gcy-22* homologs, we hypothesized that they likely represent the ASEL-type and ASER-type rGCs in *P. pacificus.*

To determine if these *gcy-7* and *gcy-22* homologs show ASEL- and ASER-specific expression, we employed promoter fusion reporters and RNA *in situ* Hybridization Chain Reaction (HCR) [39–41]. Previously, we described the first *P. pacificus* GFP reporter with ASER-specific expression, *gcy-22.3p::gfp,* is dependent on *che-1* function [16]. We found that the *gcy-7.2p::gfp* reporter showed exclusive ASEL expression (Fig 2A-2B), while the *gcy-22.2p::gfp* reporter showed ASER neuron expression, along with expression in neurons with posterior projections resembling the RMED and RMEV motor neurons in *C. elegans* (Fig 2C-2F). Like *gcy-22.3p::gfp,* the *gcy-22.5p::gfp* reporter exhibited exclusive ASER expression (Fig 2G-2H). While fluorescent reporters (GFP/RFP) allow neurites to be visible for cell identification based on axon trajectory and dendritic endings, the advent of HCR technology provides a more accurate representation of endogenous mRNA expression of multiple genes simultaneously without knowing their regulatory regions. Using HCR, we found that *gcy-7.2* expression co-localizes with both *gcy-7.1* and *gcy-7.3* transcripts in the ASEL neuron (n = 25; n = 24 respectively), whose expression are dependent on *che-1* (n = 51; n = 50 respectively) (Fig 3A-3H). Likewise, *gcy-22.3* and *gcy-22.5* transcripts are always detected together the ASER neuron (n = 40), which are also dependent on *che-1* (n = 50, both) (Fig 3I-3L). In addition, *gcy-22.3* co-localizes with *gcy-22.1* transcripts while *gcy-22.5* co-localizes with *gcy-22.2* transcripts

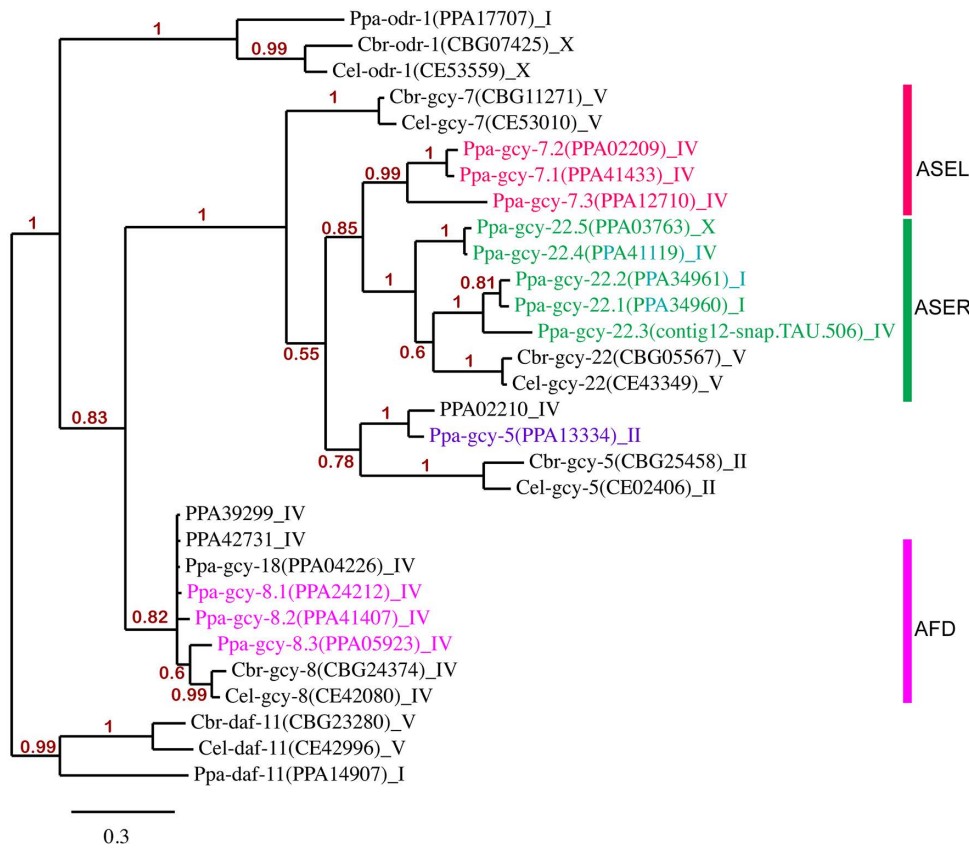

**Fig 1. Phylogeny of receptor-type guanylyl cyclases (rGCYs) in *P. pacificus* (Ppa) and *Caenorhabditis* species [*C. elegans* (Cel) and *C. briggsae* (Cbr)].** 30 full length amino acid sequences were used to perform Bayesian analysis. Genes are listed alongside transcript names and chromosome locations. Red, green, and magenta-colored genes highlight the 11 *P. pacificus* rGCYs examined in this study that are expressed in the ASEL, ASER, and AFD neurons, respectively. Five rGCYs, one in the *gcy-8* subfamily (PPA414935) and four not belonging to either of the *gcy-8* and *gcy-7*/*gcy-22* subfamilies (PPA26217, PPA38630, PPA15996, PPA22592) were not included to simplify the tree topology. The *daf-11* orthologs were designated as the outgroup. Only branch support above 0.5 are shown. The scale bar indicates genetic change. Amino acid alignment and phylogenetic analysis was performed on http://www.phylogeny.fr.

(S1 Fig). The expression level for *gcy-22.1, gcy-22.2,* and *gcy-22.4* are significantly lower than *gcy-22.3* and *gcy-22.5* as determined by co-staining experiments (S1 Fig). It is important to note that the strong *gcy-22.2p::gfp* reporter expression in RMED and RMEV neurons was not supported by HCR-FISH. Next, we confirmed that *gcy-7.2* is co-expressed with *che-1p::gfp* and is dependent on *che-1* for expression in the ASEL neuron (Fig 4A-4C). Using HCR-FISH, we found a few animals show detectable *gcy-5* transcript co-localization with *che-1p::gfp* in at least one ASE neuron, although the signal level was too weak to allow further determination of ASE laterality, if any (n = 6) (S2 Fig). In summary, transcripts of all three *gcy-7* paralogs (*gcy-7.1, gcy-7.2, gcy-7.3*) are detected exclusively in the ASEL neuron, while all five *gcy-22* homologs are expressed exclusively (*gcy-22.1, gcy-22.2, gcy-22.3, gcy-22.4, gcy-22.5*) in the ASER neuron.

## The expression of three AFD-specific *gcy-8* paralogs is dependent on CHE-1

To identify terminal effector that are selectively expressed in the *P. pacificus* AFD neurons, which could also be regulated by *che-1* [16], we next examined the expression profiles of *gcy-8*-like genes for possible conserved function. In *C. elegans*, *gcy-8, gcy-18, gcy-23* are expressed specifically in the AFD neurons and are necessary for thermosensation [42,43]. In the

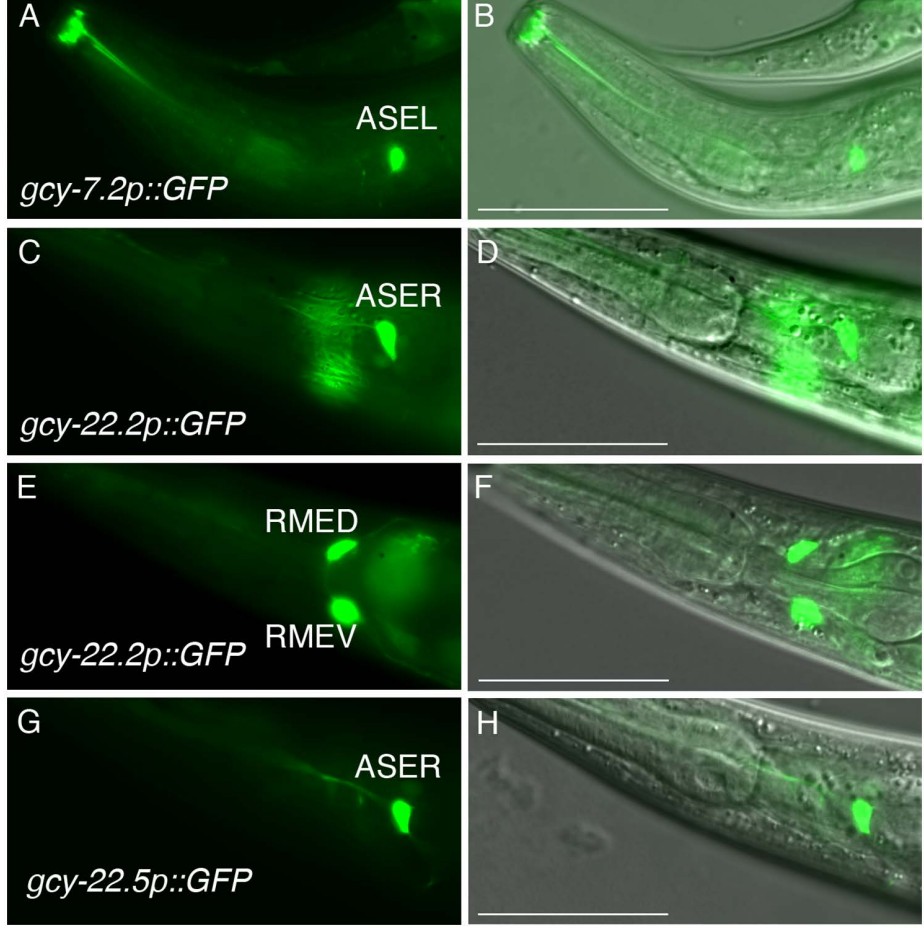

**Fig 2. P. pacificus GFP reporter expression in ASEL and ASER neurons in adult hermaphrodites.** (A-B) GFP and DIC overlay of *gcy-7.2p::GFP* expression in the ASEL neuron. (C-D) *gcy-22.2p::GFP* expression in the ASER neuron. (E-F) Same animal as panels C-D showing *gcy-22.2p::GFP* expression in putative the RMED and RMEV motor neurons based on their positioning and posterior processes running along the dorsal and ventral sides of the body. (G-H) *gcy-22.5p::GFP* expression in the ASER neuron. Anterior is left and ventral is down. The scale bars in the overlay images represent 50 μm for all panels.

human parasite *Strongyloides stercoralis*, three *gcy-23* paralogs represent the putative AFD-rGCs, with *Sr-gcy-23.2* confirmed to be specifically expressed in the AFD neurons [44]. In *P. pacificus*, there are 6 *gcy-8*-like paralogs found on Chromosome IV with very similar amino acid sequences though none represents reciprocal best-BLAST hits to the 3 AFD-specific rGCs in *C. elegans* (Fig 1). Similar to the ASE-specific rGCs, these 6 *gcy-8*-like paralogs likely represent an AFD-rGC subfamily diversification in the *Pristionchus* lineage. We found *gcy-8.1(PPA24212), gcy-8.2(PPA41407)*, and *gcy-8.3(PPA05923)* transcripts to be exclusively and robustly expressed in the AFD neurons that co-express *che-1p::gfp* (Fig 4D-4L).

To determine if the expression of *gcy* genes in ASE neurons is dependent on CHE-1 function, we next examined the *gcy* genes with strong ASE expression in *che-1(-)* animals. Like the *gcy-22.3p::gfp* reporter, we found that endogenous mRNA expression of *gcy-22.3* and *gcy-22.5*, as well as *gcy-7.1, gcy-7.2*, and *gcy-7.3* were completely lost in *che-1(-)* animals (Table 1) (Figs 3, 4). In comparison, the loss of *che-1* resulted in a significant reduction of the expression the 3 *gcy-8* paralogs in larval and adult animals. Compared to 100% of wild-type animals having expression of all 3 *gcy-8* paralogs in both AFD neurons, only 28% and 24% of *che-1* animals express *gcy-8.1* and *gcy-8.2* in both AFD neurons, respectively

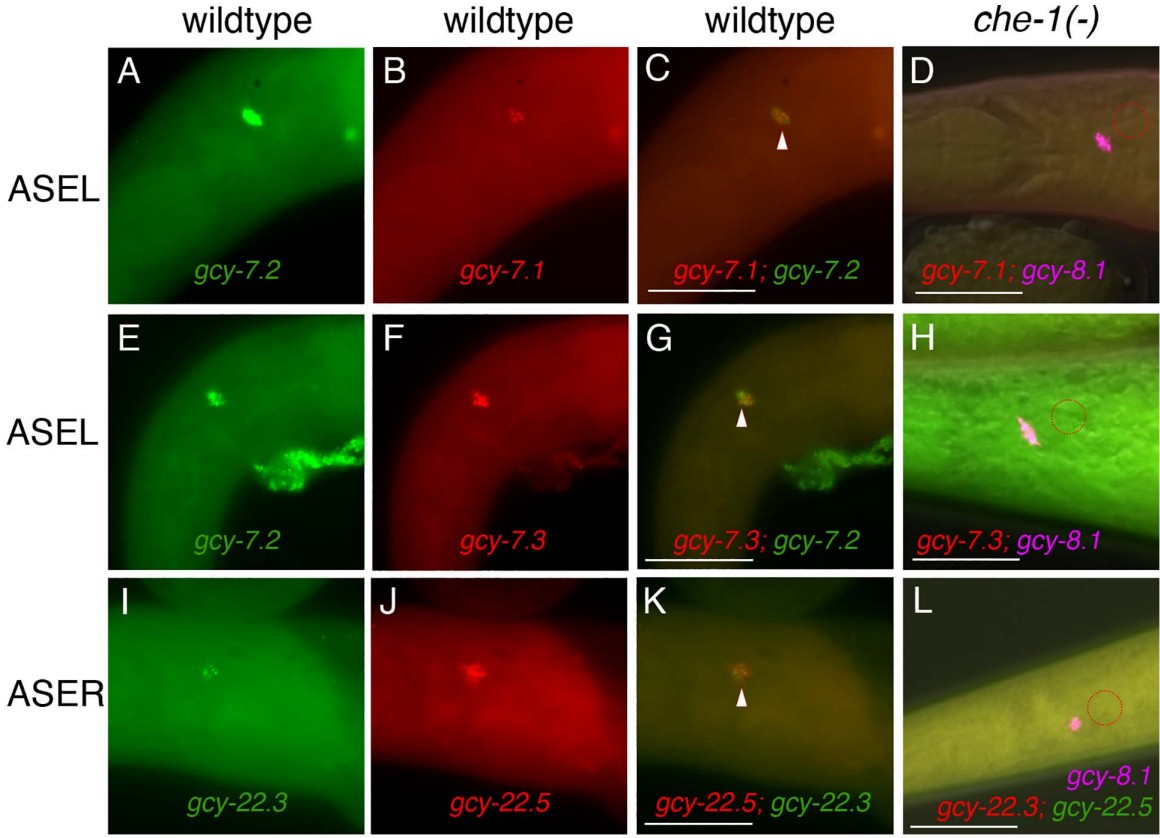

**Fig 3. Co-expression of multiple *gcy* transcripts in ASEL and ASER neurons are dependent on *che-1* expression.** Triangles in wild-type J4 and adult hermaphrodites indicate overlapping expression. (A-C) *gcy-7.1* and *gcy-7.2* overlap in ASEL. (D) *gcy-7.1* is absent in *che-1(-)* animals (n = 51). (E-G) *gcy-7.3* and *gcy-7.2* overlap in ASEL. (H) *gcy-7.3* is absent in *che-1(-)* animals (n = 52). (I-K) *gcy-22.3* and *gcy-22.5* overlap in ASER but (L) both transcripts are absent in *che-1(-)* animals (n = 40). Dotted circles denote missing expression of respective ASE markers. *che-1(-)* samples have at least one neuron with *gcy-8.1* staining. The scale bar represents 25 µm. (See Table 1 for details).

(Table 2). Moreover, only 10% of the *che-1* animals express *gcy-8.3* in both AFD neurons. The degree of reduction of *gcy-8* expression in *che-1* mutant animals is likely an underestimation, since only animals with at least one AFD neuron staining were scored but those without any AFD expression were not included in the overall count. In total, we have comprehensively identified in *P. pacificus* two ASE-rGC subfamilies comprised of 8 *gcy* genes, at least 5 of which depend on CHE-1 for their expression (expression of *gcy-22.1*, *gcy-22.2*, *gcy-22.4* were too weak to ascertain *che-1*-dependence). We also identified that at least 3 of the 6 AFD-rGC subfamily members show AFD-specific expression that require CHE-1 function for wild-type level expression.

### Embryonic AFD neurons show bipotential fates of asymmetric ASE neurons

Next, we proceeded to adopt 3 *gcy* genes as markers in *P. pacificus* for differentiated state to determine when lateral asymmetry is specified during neuronal development (*i.e.,* *gcy-7.2* for ASEL-type, *gcy-22.3* for ASER-type, and *gcy-8.1* for AFD-type). In wild-type adult hermaphrodites, *gcy-7.2* and *gcy-22.3* are invariantly expressed in the ASEL and ASER neurons, respectively, and neither overlaps with *gcy-8.1* expression (n = 51). *Gcy-7.2* expression is also limited to a single ASE neuron in J2 and J3 stage larvae that co-localize with the *che-1p::gfp* reporter (n = 15), as well as in the pre-hatching J1 larvae found characteristically in Diplogastrid species (Fig 5; 19 out of 20 J1, Table 3) [45,46]. Surprisingly, we detect the

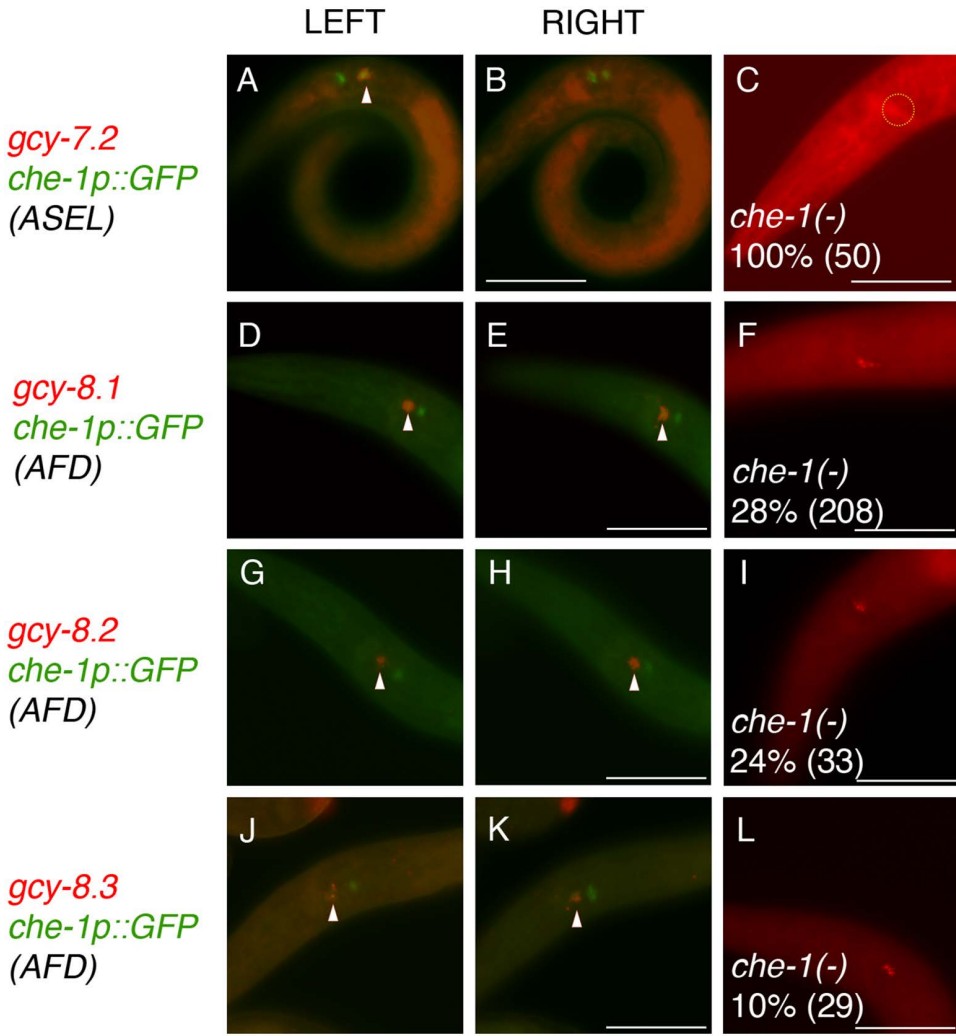

**Fig 4. gcy transcripts found in che-1-expressing ASEL and AFD neurons are dependent on che-1 expression.** Triangles in wild-type J2 and J3 larvae indicate overlapping expression. (A-B) *gcy-7.2* and *che-1p::GFP* expression in ASEL. (D-E) *gcy-8.1* and *che-1p::GFP* expression in AFD. (G-H) *gcy-8.2* and *che-1p::GFP* expression in AFD. (J-K) *gcy-8.3* and *che-1p::GFP* expression in AFD. (C) *gcy-7.2* expression is absent in *che-1* mutants (dotted circle). (F) *gcy-8.1* expression is present in both AFD neurons in 28% of *che-1* animals. (I) *gcy-8.2* expression is present in both AFD neurons in 24% of *che-1* animals. (L) *gcy-8.3* expression is present in both AFD neurons in only 10% of *che-1* animals. The scale bar represents 25 μm. See Tables 1 and 2 for details.

earliest expression of *gcy-7.2* and *gcy-22.3* occurs in late-stage embryos (~3-fold), with up to 3 neurons expressing *gcy-7.2* or *gcy-22.3* (Fig 5; n = 54, Table 4). Specifically, of the 2 neurons that express the AFD marker, *gcy-8.1*, one or both neurons also co-stain for the ASE markers. That is, *gcy-7.2* and *gcy-22.3* expression overlap only in late embryogenesis in 1–2 neurons and always with the AFD marker, *gcy-8.1*. Hence, a hybrid state expressing all 3 CHE-1-dependent markers for ASEL, ASER and AFD exist in the AFD neurons prior to the J1 larval stage.

### A DIE-1 homolog regulates ASE left-right asymmetry important for salt chemosensory behavior

To identify genes that are required to establish asymmetry in *P. pacificus*, we performed a non-saturating forward genetic screen for Defects in ASE ASymmetry (*das*). Specifically, we looked for mutants that expressed the *gcy-22.3p::gfp*

**Table 1. ASE-rGC (number of animals with a single ASE neuron stained).**

|  | gcy-7.1 | gcy-7.2 | gcy-7.3 | gcy-22.3 | gcy-22.5 |
|---|---|---|---|---|---|
|  | ASEL | ASEL | ASEL | ASER | ASER |
| Wild type (J2-Ad) | 100% (51) | 100%(>150) | 100% (52) | 100%(>150) | 100% (40) |
| Wild type (J1) | 100% (3) | 100% (22) | 100% (4) | 100% (16) | 100% (21) |
| che-1(ot5012) (J2-Ad)* | 0% (51) | 0% (50) | 0% (50) | 0% (50) | 0% (50) |

*Result from che-1(-) samples with at least one AFD neuron with gcy-8.1 staining.

**Table 2. AFD-rGC (number of animals with two AFD neurons stained).**

|  | gcy-8.1 |  | gcy-8.2 |  | gcy-8.3 |  |
|---|---|---|---|---|---|---|
| 2xAFD stained | larva/adult | embryo | larva/adult | embryo | larva/adult | embryo |
| Wild type | 100% (126) | 100% (59) | 100% (135) | 100% (13) | 100% (129) | 100% (27) |
| che-1(ot5012)* | 28%(208) | 52% (50) | 24% (33) | 50% (6) | 10% (29) | 17% (12) |
| ttx-1; che-1** | 0%** (50) | ND | ND | ND | ND | ND |

*From 2 independent staining experiments from samples with at least one AFD neuron staining.

**From samples with intestinal Ppa-hsd-2 staining.

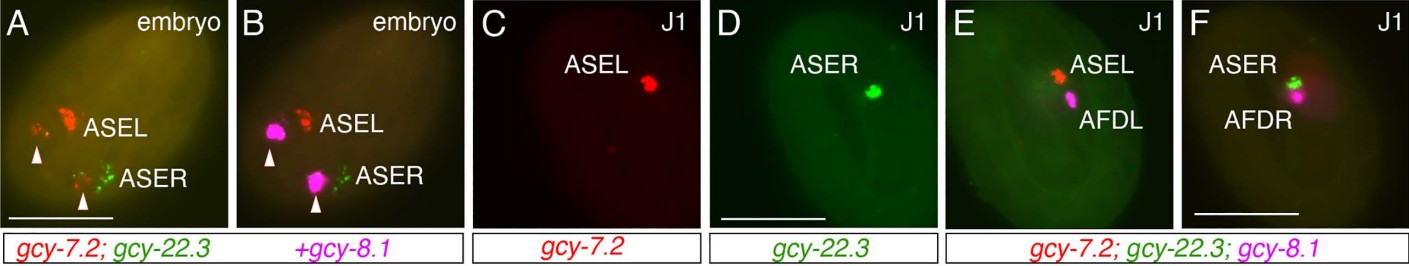

**Fig 5. AFD neurons exhibit ASE differentiation marker in embryos but not in J1 larvae.** (A-B) A late-stage embryo with gcy-7.2 and gcy-22.3 expression in AFD neurons (Triangles)(53%; n = 54). (C-F) A pre-hatching J1 larva shows only gcy-7.2 or gcy-22.3 expression in ASE neurons, and gcy-8.1 expression in AFD neurons (95%; n = 20). Scale bar represents 25 µm. See Table 3 for details.

reporter in both ASE neurons, targeting genes needed to repress the ASER fate in the ASEL neuron (*i.e.,* 2xASER Class II mutant phenotype) [36]. We identified a fully penetrant *das* allele (100%, n = 50, Fig 6A), *das-1(csu222)*, which contained a GCT>ACT (A359T) substitution in *PPA12810*, a homolog of the *Cel-die-1* that encodes a zinc-finger transcription factor required for establishing lateral asymmetry in the ASE neurons [26,27,35,36]. The *csu222* point mutation is in the second Zn-finger domain of Ppa-DIE-1. To confirm that this *Ppa-die-1* mutation is solely responsible for the *das* phenotype, we used CRISPR/Cas9 to reproduce an identical mutation in a wild-type background. When this genome-edited line was then crossed into the unmutagenized *gcy-22.3p::gfp* reporter line, we found that *die-1(csu225)* fully phenocopied the 2xASER *das-1(csu222)* phenotype (100%, n = 40, Fig 6B). Thus, we have identified through an unbiased genetic screen the *Ppa-die-1* homolog required for establishing lateral asymmetry in the ASE neurons in *P. pacificus*.

We next examined in detail the effect of reduction-of-function *die-1* on other neuronal markers. Surprisingly, at least two of the ASER-specific *gcy-22* paralogs are affected differently by the *die-1* mutation. While both *das-1(csu222)* and its

**Table 3. Expression phenotypes of *Ppa-die-1* mutants. HCR using *gcy-8.1*(B5-magenta):AFD, *gcy-7.2* (B4-red):ASEL, *gcy-22.3* (B2-green):ASER.**

| | 1 ASEL, 1 ASER | 1 ASEL, 2 ASER | 2 ASEL, 2 ASER | 2 ASEL, 2 ASER | 3 ASEL, 2 ASER | 3 ASEL, 1 ASER | 3 ASEL, 2 ASER |
|---|---|---|---|---|---|---|---|
| Wild type J2-Ad (51) | 100% (51) | — | — | — | — | — | — |
| Wild type J1 (20) | 95% (19) | — | — | — | — | 5% (1) | — |
| Wild type embryos (54) | 47.2% (25) | — | — | — | — | 38.9% (21) | 15% (8) |
| *die-1(csu225)* J2-Ad (52) | — | 73.1% (38) | 9.6% (5) | 6% (3) | 11.5% (6) | — | — |
| *die-1(csu225)* J1 (11) | — | 54.5% (6) | 9.1% (1) | 27.2% (3) | 9.1% (1) | — | — |
| *die-1(csu225)* embryos (31) | 6.2% (2) | 6.2% (2) | — | 3.1% (1) | 16.1% (5) | 61.3% (19)* | 6.2% (2) |

The most frequent category is highlighted in red.

*5 of the 19 *die-1* embryos have only one AFD with *gcy-7.2* misexpression.

**Table 4. Comparison of embryonic vs post-embryonic *gcy* gene expression in various mutants that affect AFD expression. HCR-FISH using *gcy-8.1*(B5-magenta):AFD, *gcy-7.2* (B4-red):ASEL, *gcy-22.3* (B2-green):ASER.**

| Category | wild type | 1 | 2 | 3 | 4 | 5 | Other |
|---|---|---|---|---|---|---|---|
| | 1 ASEL, 1 ASER | 3 ASEL, 2 ASER | 3 ASEL, 1 ASER | 1 ASEL, 3 ASER | 3 ASEL, 3 ASER | 0 ASEL, 0 ASER | |
| Wild type (PS312) J2-Ad (51) | 100% (51) | — | — | — | — | — | — |
| embryos (54) | 47.2% (25) | 7.5% (4) | 38.9%* (21) | — | 7.5% (4) | — | — |
| *das-2(csu223)* J2-Ad (50) | 24% (12) | — | 8% (4) | 22% (11) | 46% (23)** | — | — |
| *ttx-1(csu150)* J2-Ad (50) | 24.5% (12) | — | 20.4% (10) | — | 32% (16) | 14.3% (7) | 10.2% (5) |
| *ttx-1(csu150)* embryos (32) | 6.2% (2) | — | — | — | 84.4% (27) | — | 9.3% (3) |
| *tax-4(cbh68)* J2-Ad (53) | 30.2% (16) | — | 64.1% (34) | — | — | — | 5.7% (3) |
| *tax-4* embryos (32) | 6.2% (2) | 18.8% (6) | 12.5% (4) | — | 40.6% (13) | 9.4% (3)**** | 2.5% (4) |
| *tax-2(cbh46); tax-4(cbh68)* J2-Ad (51) | 5.9% (3) | — | 82.4% (42) | — | — | 7.8% (4) | 3.9% (2) |
| *tax-2(cbh46); tax-4(cbh68)* embryos (32) | 9.3% (3) | 9.3% (3)*** | 18.8% (6) | — | 59.4% (19) | 3.1% (1)**** | — |

Wild type is shown for comparison. The most frequent category is highlighted in red.

*15 of the 21 samples have only 1 AFD with both *gcy-7.2* expression.

**12 of 23 samples with the *gcy-8.1, gcy-7.2. gcy-22.3*-expressing AFD neurons (52%) in *das-2* have an extra *gcy-22.3*-expressing neuron posterior to the ASER.

***One AFD with both *gcy-7.2* and *gcy-22.3* expression, and another AFD with *gcy-7.2* or *gcy-22.3* or neither.

****gcy-7.2 present in the ASEL but no *gcy-22.3* expression in the ASER.

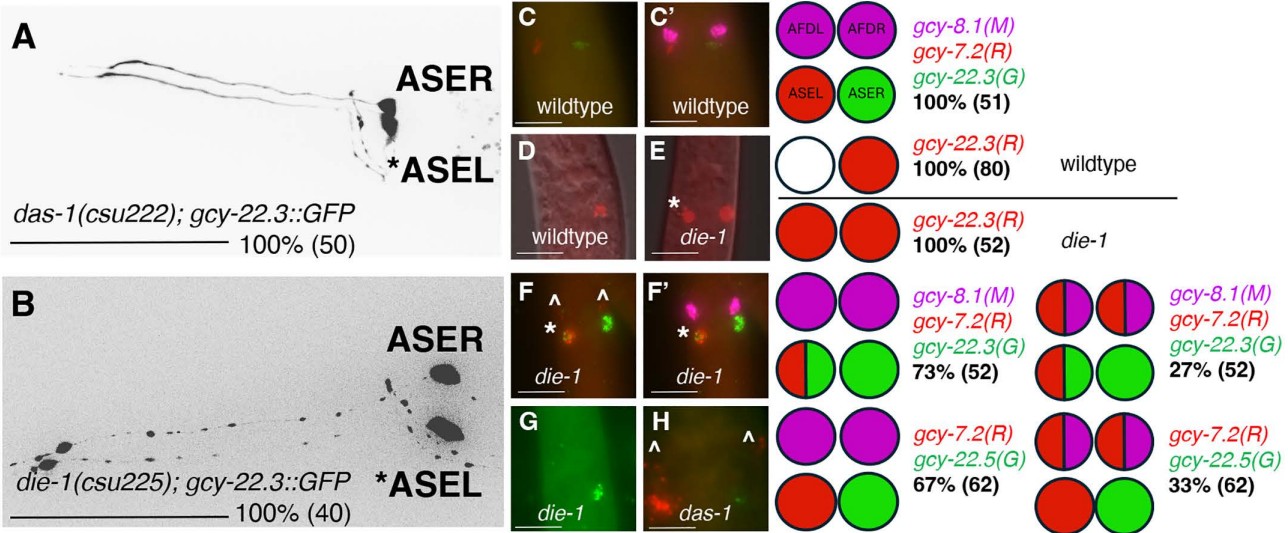

**Fig 6. *Ppa-die-1* mutation results in *gcy* misexpression in ASEL and AFD neurons.** (A-B) Max projection images of adults with the *gcy-22.3p::gfp* reporter. *ASEL shows misexpression of the *gcy-22.3* ASER marker. (A) Forward genetic screen identified *das-1(csu222)* based on its "2xASER" mutant phenotype. (B) CRISPR/Cas9-edited *die-1(csu225)* phenocopies the "2xASER" mutant phenotype. (C). HCR-FISH showing wild-type expression pattern of *gcy-7.2(red)* and *gcy-22.3(green)* in ASEL and ASER neurons. (C') Sample in (C) overlaid with *gcy-8.1(magenta)* staining. (D-E) Wild type compared to the *die-1* "2xASER*" phenotype using *gcy-22.3(B4)* labeled with a red fluorophore. Misexpression of *gcy-22.3(B4)* in the ASEL neuron is marked by *. (F) *die-1(csu225)* shows *gcy-22.3(B2)* misexpression in the ASEL neuron (*) and weak *gcy-7.2(B4)* misexpression in both AFD neurons (below ^). (F') Sample in (F) overlaid with *gcy-8.1(B5)* staining. (G) *die-1(csu225)* does not show misexpression of another ASER marker, *gcy-22.5(B2)*. (H) Co-staining of *das-1(csu222)* with *gcy-7.2(B4)* and *gcy-22.5(B2)* shows *gcy-7.2(B4)* misexpression in AFD neurons, consistent with the *die-1(csu225)* staining. Sample sizes are indicated in parentheses. Scale bars represent 50 µm in (A-B) and 5 µm in (C-H).

phenocopy, *die-1(csu225)*, express two *gcy-22.3p::gfp* ASE neurons (Fig 6), *gcy-22.5p::gfp* is only expressed in a single ASER neuron (*i.e.,* wild type-like). The two ASER marker expression in *die-1* mutants is corroborated by HCR-FISH, with *gcy-22.3* staining in both ASE neurons (Table 3), versus *gcy-22.5* staining only in the ASER neuron. This shared difference in the two ASER markers further confirm that *das-1(csu222)* and *die-1(csu225)* are phenotypically equivalent. Intriguingly only ~26% (out of 31) of *die-1(csu225)* embryos exhibited the 2xASER phenotype compared to 100% of the *die-1(csu225)* larvae and adults, which suggests DIE-1 acts post-embryonically to maintain repression of ASER fate in the ASEL. Interestingly, when we examined the expression of the ASEL marker *gcy-7.2*, 27% of the *die-1* larval and adult animals showed weak ectopic *gcy-7.2* expression in one or both AFD neurons as determined by *gcy-8.1* co-staining (n = 52) (Fig 6). Lastly, we found that *die-1* is expressed in many cell types, including the ASEL, but specifically absent in the ASER as would be expected for its normal role in repressing ASER fate in the ASEL neurons (n = 45)(S3 Fig). In summary, post-embryonic *die-1* mutant animals have an ASEL neuron with both ASEL and ASER attributes, as well as AFD neurons with ASEL attributes. We designate this class of phenotype as "2xASER*."

To assess whether neuronal fate changes affect function, we tested if *die-1* mutants altered their response to attractive salts in chemotaxis assays. We previously found the loss of *gcy-22.3* had no detectable effect on attraction to the same panel of salts [16]. We found that compared to wildtype, *Ppa-die-1* mutants exhibited enhanced attraction to various ions- $NH_4^+$, $Cl^-$, and $Na^+$ (with a salt background), as well as enhanced attraction to $NH_4Cl$ and NaCl (without a salt background) (Fig 7). In contrast, *die-1* mutants did not exhibit the observed enhanced attraction towards iodide ions ($I^-$) nor $NH_4I$. In particular, attraction to $Cl^-$ ($NH_4Cl$ on $NH_4I$ background), was completely abolished in wildtype, but significantly less affected in *die-1* mutants. Thus, the misexpression of *gcy-22.3* in both ASE neurons and *gcy-7.2* in both AFD neurons, possibly along with other asymmetrically expressed ASE-type *gcy* genes (although not *gcy-22.5*), significantly

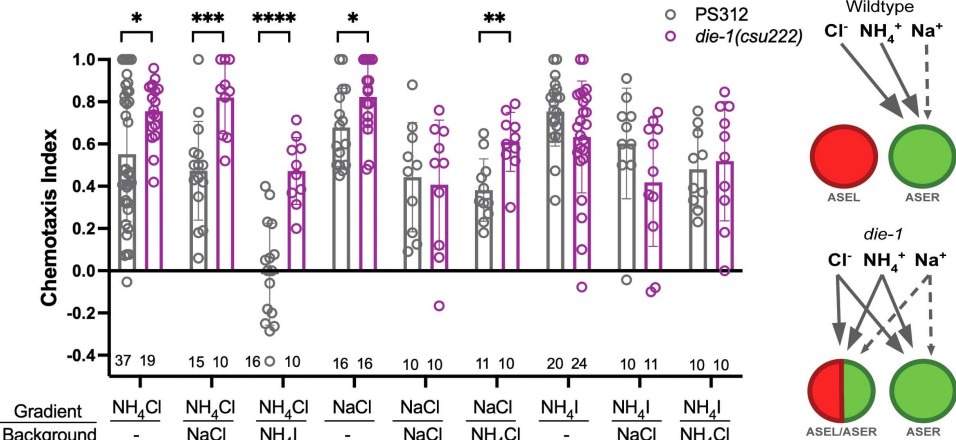

**Fig 7.** *die-1* **mutants show enhanced responses to specific salt ions.** J4 to adult hermaphrodite responses to $NH_4^+$, $Na^+$, $Cl^-$, and $I^-$ ions in the presence of $NH_4Cl$, $NaCl$, $NH_4I$ or no salt (-) background. *P<0.05, **P<0.01, ***P<0.001, ****P<0.0001 significant differences were found between wild type PS312 and *das-1/die-1(csu222)* for $NH_4Cl$ on all backgrounds tested, and for $NaCl$ on $NH_4Cl$ background by unpaired t-test and Mann-Whitney test. Sample sizes for each condition are indicated on the bottom. The proposed scenario of ions detected by the ASEL and ASER neurons in wild type versus *die-1* mutant animals ($Cl^-$, $NH_4^+$, $Na^+$). Solid versus dashed arrows indicate relative sensitivity of neurons for sensing each ion.

altered chemosensory behavior. Thus, from both the prior calcium imaging experiments and these chemotaxis assays, we conclude that sensitivity to $NH_4^+$, $Cl^-$, and $Na^+$ — but not $I^-$ — are likely associated with misexpression of terminal identity genes normally expressed in a single ASE neuron.

## CHE-1, TTX-1, TAX-2, and TAX-4 regulate AFD identity

In addition to *das-1*, we also isolated a *das-2(csu223)* allele that represented another class of mutant ASER phenotype, which showed ectopic expression of *gcy-22.3p::gfp* in a pair of amphid neurons anterior to the ASEs that could be the AFDs (S4 Fig). We further examined the *das-2* phenotype using the ASE and AFD markers by HCR-FISH (*gcy-7.2, gcy-22.3, gcy-8.1*) and discovered that not only was *gcy-22.3* ectopically expressed in one or both AFD neurons (68%, 34 out of 50), *gcy-7.2* was also ectopically expressed in one or both AFD neurons (54%). Notably unique to the *das-2* phenotype are animals that ectopically express *gcy-22.3* but not *gcy-7.2* in AFD neurons (22%, Table 4). Furthermore, we also detected *gcy-22.3* expression in an unidentified neuron ventral and posterior to ASER in 24% of *das-2* mutant post-embryonic animals. Because the OTX homeodomain transcription factor CEH-36 is expressed in ASE and AWC neurons and responsible for specifying ASE fate in *C. elegans* [19,36,47], we wondered if the putative *P. pacificus* homolog, *Ppa-ceh-36.1*, is linked to the *das-2* phenotype. However, we did not detect any mutations in the ~3.7 Kb genomic region of *Ppa-ceh-36.1(PPA27298)* in *das-2*, including the putative promoter and 3' UTR. The *das-2* allele remains unclonced because of the difficulty in maintaining due to high embryonic lethality and the inability to outcross this strain. Yet, given that in *P. pacificus che-1* and the putative AFD terminal selector, *ttx-1,* are both expressed in the AFD neurons [16], we wondered if TTX-1 contributes to the negative regulation of ASE fate in the AFD neurons.

In *C. elegans, ttx-1* is the terminal selector for the thermosensory AFD neurons [48,49]. We have shown previously that both TTX-1 and CHE-1 protein and transcriptional reporters co-localize in the AFD neurons, suggesting they likely interact genetically [16]. To determine the role of Ppa-TTX-1 in specifying AFD identity, we generated a viable reduction-of-function allele in *P. pacificus* (Fig 8A). We could not recover homozygous alleles for loss-of-function mutations, such as heterozygote carriers of frame-shift mutations, likely because *Ppa-ttx-1* is also required for embryogenesis. To determine if the reduction-of-function *Ppa-ttx-1* affects thermosensation, we compared wildtype versus *Ppa-ttx-1(csu150)* and found that

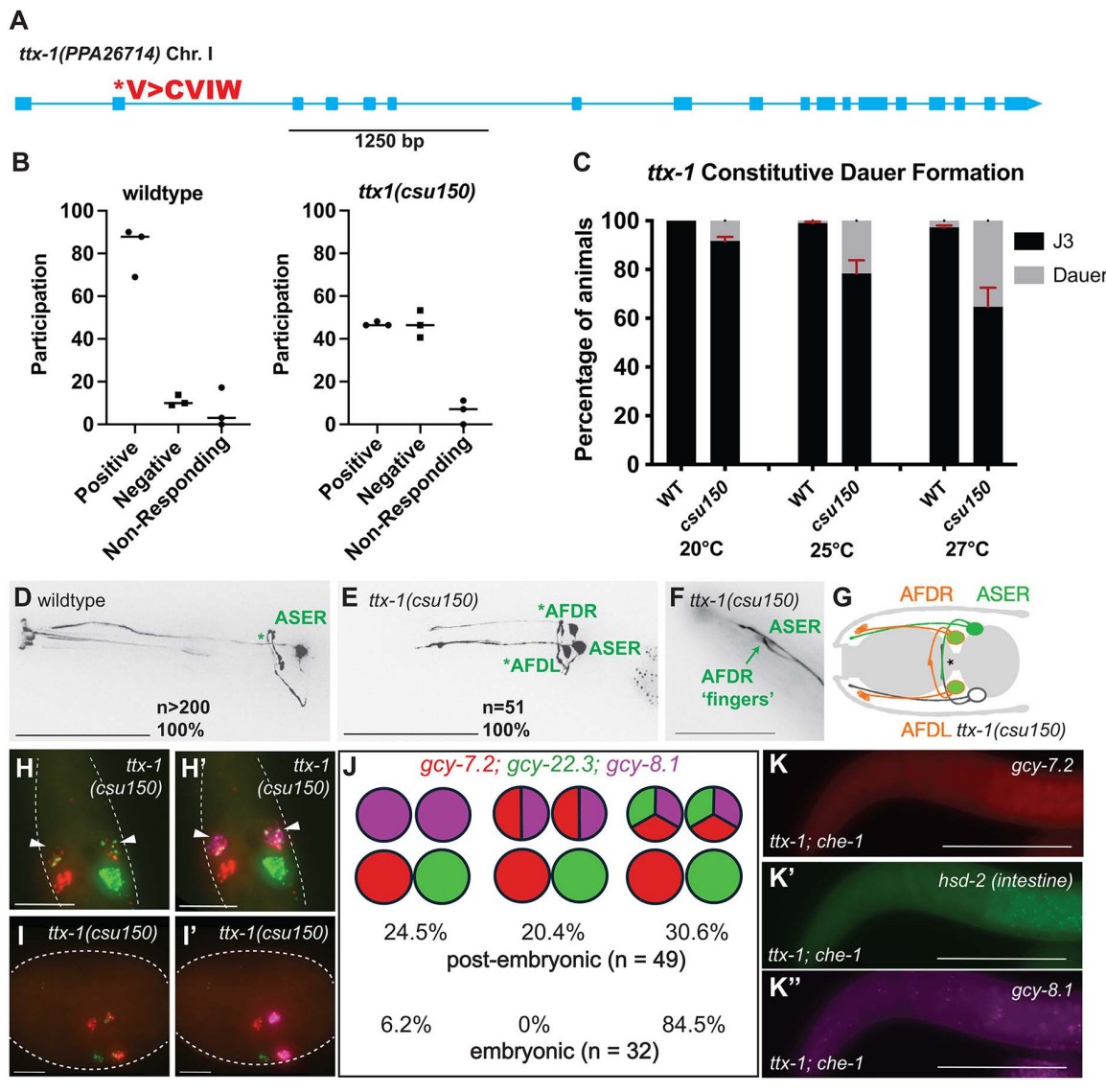

**Fig 8. *Ppa-ttx-1* reduction-of-function mutant phenotypes.** (A) PPA26714 on Chromosome I is the best reciprocal BLAST ortholog of *ttx-1* (Pristionchus.org: "El Paco V3"). The predicted second exon likely contains the start codon, where 9 bp is inserted in frame in *ttx-1(csu150)*(V>CVIW). (B) In the thermotaxis assay, worms cultured at 20°C were loaded at 25°C on a temperature gradient from 22°C-34°C and tracked by time lapse recording (3 independent trials per genotype). Y axis is percentage of worms participating in thermotaxis. Positive thermotaxis means worms moved towards higher temperature (>25°C), and negative thermotaxis means worms moved towards the lower temperature (<25°C). A majority of wild-type *P. pacificus* showed positive thermotaxis while *ttx-1(csu150)* mutants showed no preference. (C) In the Daf-c dauer formation assay, worms were cultured for 4 days and scored with food at 20°C, 25°C, and 27°C (100-150 animals per assay; 6 assays for each condition). Constitutive dauer formation in *ttx-1(csu150)* was positively correlated with higher temperature. Error bars denote 95% confidence interval. (D) Maximum projection of *gcy-22.3p::gfp* in a wild type adult shows the characteristic ASE axon that passes over the dorsal midline and its own dendrite (marked by *) (E). Maximum projection of *gcy-22.3p::gfp* ectopic expression in both AFD neurons in a *ttx-1(csu150)* mutant. (F) AFD dendritic end shows wild-type morphology in *ttx-1(csu150)* mutant. (G) Schematic summary of *gcy-22.3p::gfp* expression in the *ttx-1(csu150)* mutant. (H). HCR-FISH of a *ttx-1(csu150)* J3 larva showing ectopic expression of *gcy-7.2(B4)* and *gcy-22.3(B2)* in AFD neurons. (H') Sample in (H) overlaid with *gcy-8.1(B5)* staining. (I). HCR-FISH of a *ttx-1(csu150)* late-stage embryo showing ectopic expression of *gcy-7.2(B4)* and *gcy-22.3(B2)* in AFD neurons. (I') Sample in (I) overlaid with *gcy-8.1(B5)* staining. (J) Summary of the *ttx-1(csu150)* phenotype with comparison of the distribution of post-embryonic (J1 to adult) versus embryonic ASE misexpression patterns. (K; K'; K") HCR-FISH of a *ttx-1; che-1* double mutant J3 larva lack expression for *gcy-7.2(B4)* and *gcy-8.1(B5)* in ASE and AFD neurons. Only samples with *hsd-2(B2)* staining in the intestine were scored (n = 50). Sample sizes are indicated in parentheses. Scale bars in (D, E, K) represent 50 µm, while the scale bars in (F-I) represent 5 µm.

while wild-type animals raised at 20°C placed at 25°C tracked towards warmer temperature, *Ppa-ttx-1(csu150)* subjected to the same conditions showed no thermal preference (Fig 8B). Moreover, *Ppa-ttx-1(csu150)* also showed a temperature-dependent constitutive dauer defect (Daf-c) akin to the dauer-pheromone hypersensitivity observed in *Cel-ttx-1(p767)* mutants (Fig 8C) [50–52]. The thermotaxis and Daf-c mutant phenotypes of *Ppa-ttx-1(csu150)* reduction-of-function allele are similar to the reference allele in *C. elegans, Cel-ttx-1(p767)* [48].

We proceeded to examine the expression pattern the terminal effector genes in the *Ppa-ttx-1(csu150)* mutant. *Ppa-ttx-1(csu150)* mutants still retain terminally differentiated AFD features– robust *gcy-8.1* expression and the distinctive finger-like dendritic endings were observed in both AFD neurons (Table 4). This finding highlights significant differences in comparison to AFD neurons in the *Cel-ttx-1(p767)* mutant, which lacks finger-like endings and *gcy-8* expression (Fig 8D-8G) [48,53]. Unexpectedly, *Ppa-ttx-1(csu150)* mutants misexpress the *gcy-22.3p::gfp* ASER reporter in both AFD neurons (100%, n = 51), as well as by HCR staining (Fig 8H-8J) (49%, n = 50). Moreover, the *gcy-7.2* ASEL marker is also ectopically expressed in 3 neurons– ASEL and both AFD neurons (weakly)– suggesting the transformation of AFD to ASE-like fate does not confer lateral asymmetry (Fig 8J). Compared to just 7.5% of wild-type embryos (out of 54), 84.4% of *ttx-1(-)* embryos (out of 32) expressed all 3 markers in the AFD neurons (Fig 8I-8J), suggesting TTX-1 contributes to the restriction of ASE-specific *gcy* expression in the AFD precursors during late embryogenesis. To address the potential that *Ppa-ttx-1* also has a role in activating AFD-specific gcy genes in a *che-1* dependent manner, we generated *ttx-1(csu150); che-1(ot5012)* double mutants and observed loss of all AFD (*gcy-8.1*) and ASEL (*gcy-7.2*) expression (Fig 8K)(0% staining, n = 50; Table 2). This complete loss of *gcy-8.1* expression is more severe than in the *che-1* mutant alone (28% staining) and suggests *ttx-1* possesses positive instructional or maintenance role in AFD terminal differentiation. Taken together, TTX-1 in *P. pacificus* amphid neurons appears to be necessary for the repression of *che-1*-dependent ASE differentiation in the AFD, while also functioning cooperatively with CHE-1 for the promotion of AFD-specific fates.

The cGMP-dependent signaling pathway is also involved in restricting the ASE fate in AFD neurons. In *C. elegans*, mutations in the cyclic nucleotide gated channels (CNG) encoded by *tax-2* and *tax-4* resulted in the ectopic expression of the ASEL marker (*lim-6p::gfp*) but not the *gcy-7p::gfp* marker in AFD neurons ("Type VI" mutants) [36]. To determine if *P. pacificus* CNGs have a similar role in neuronal patterning during development, we examined *P. pacificus tax-4(cbh68)* single and *tax-2; tax-4 (cbh46; cbh68)* double mutants, which were previously shown to have defects in light avoidance [54]. In *P. pacificus*, transcriptional fusion reporters of both *tax-2* and *tax-4* are expressed in 8 out of 12 amphid neurons, including the AFD and ASE neurons [54]. We found that mutants in *tax-4(cbh68)* and *tax-2; tax-4 (cbh46; cbh68)* displayed ectopic *gcy-7.2* expression in one or both AFD neurons, with *tax-2; tax-4* larvae and adults exhibiting a stronger phenotype than *tax-4* single mutants (82.4% versus 64.1%)(Fig 9 and Table 4). We observed a small percentage of *tax-2; tax-4* post-embryonic animals (7.8%), as well as *tax-4* and *tax-2; tax-4* embryos (9.4% and 3.1%, respectively) lacked *gcy-22.3* expression. *Ttx-1(csu150)* post-embryonic animals also lacked expression in both ASE markers (14.3%), suggesting *ttx-1, tax-2,* and *tax-4* may also have a minor instructive role in specifying ASE fate (Table 4). The similar phenotypes of *ttx-1* and *tax-2; tax-4* mutants indicate they repress ASE fate in AFD neurons post-embryonically.

## Multiple miRNAs are implicated in the regulation of ASE and AFD fates

In *C. elegans,* DIE-1 acts through the *lsy-6* miRNA in repressing *cog-1* expression to regulate neuronal cell fates in ASER and ASEL. Since it is unclear if a functional *lsy-6* miRNA homolog exists in the *Pristionchus* lineage [29], we wondered if other miRNAs had independently evolved to fulfill the demands of the regulatory network for establishing ASE lateral asymmetry. First, to determine if miRNAs are involved in establishing ASE asymmetry in *P. pacificus*, we disrupted the function of *Ppa*-PASH-1, an evolutionarily conserved enzyme required for the biogenesis of mature miRNAs. We identified the putative *Ppa-pash-1* homolog and engineered a substitution mutation (S494Y) in the same mutated residue as the viable temperature-sensitive allele in *Cel-pash-1(mj100)*. To confirm the molecular impact of the *pash-1* reduction-of-function allele, we performed small RNAseq and found *Ppa-pash-1* has a significantly lower percentage of reference miRNAs and

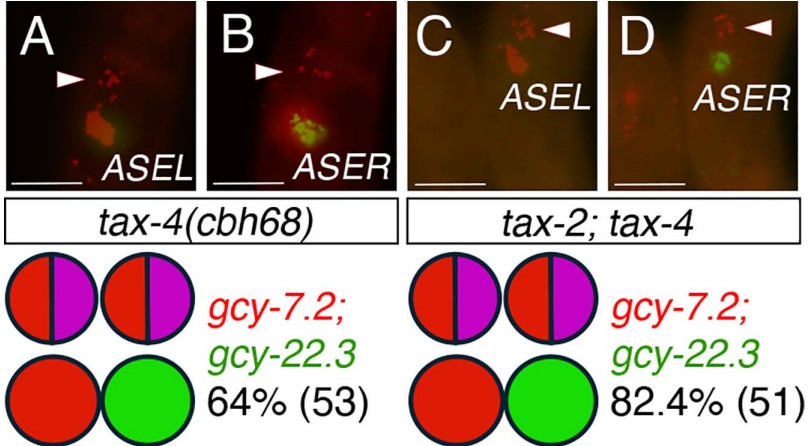

**Fig 9. Cyclic nucleotide gated channels TAX-2 and TAX-4 contribute to AFD fate.** HCR-FISH with *gcy-7.2* for ASEL fate and *gcy-22.3* for ASER fate. (A-B) *tax-4(cbh68)* J3 larva with *gcy-7.2* misexpression in AFD neurons (Triangles). (C-D) *tax-2(cbh46); tax-4(cbh68)* J3 larva with *gcy-7.2* misexpression in AFD neurons. The pictogram below each genotype indicates the most frequent expression phenotype with the total sample size in parenthesis. Scale bars represent 5 μm. See Table 4 for details.

more sequence variants compared to wildtype (S5 Fig and S1 Appendix). Thus, the *Ppa-pash-1(csu227)* allele resulted in a genome-wide loss of correctly-processed mature miRNAs. We observed that late embryonic *Ppa-pash-1(csu227)* mutants show a complete transformation of the ASEL to ASER fate (n = 32) (Fig 10A). Similarly, larval *Ppa-pash-1* mutants also showed a complete penetrance of the 2xASER phenotype, albeit with noticeable minority also exhibiting a hybrid ASER/AFD phenotype not seen in any other mutant classes (7.4%, n = 54)(Fig 10B). The misexpression of *gcy-22.3* in both ASE neurons in *Ppa-pash-1(csu227)* mutants was invariant during late embryogenesis (*e.g.,* both ASEs express only *gcy-22.3* but not *gcy-7.2*), in contrast to 100% of wild-type embryos that showed only one ASEL and one ASER marker in neurons that lack *gcy-8.1* co-expression for AFD fate (n = 54)(Tables 4 and 5). Thus, the conserved *Ppa*-PASH-1 function for the processing of mature miRNAs is necessary to repress ASER-fate in the ASEL and the AFD neurons primarily during the post-embryonic stages.

Since in *C. elegans lsy-6* miRNA repression of *cog-1* is necessary for the ASER fate repression in the ASEL neuron, we hypothesized that the *P. pacificus cog-1* homolog is also down-regulated post-transcriptionally in an asymmetric manner. We utilized the *Ppa-che-1* promoter containing the first exon and intron to drive expression of *gfp* in the ASE and AFD neurons (*Ppa-che-1p::gfp:cog-1 3' UTR*) and compared ASE expression to the well-characterized constitutive expression of the *Ppa-che-1p::gfp:rpl-23 3' UTR* transgene [55]. Using AFD expression as a control for transgene expression, we found that the *Ppa-cog-1 3'UTR* resulted in animals with significantly reduced ASEL expression versus ASER (20 out of 63) compared to the reporter with the constitutive *rpl-23 3'-UTR* (0 out of 54, Fisher's Exact test P < 0.0001)(Fig 10). Thus, the 3'-UTR is sufficient to confer reduction of GFP expression asymmetrically in the ASEL neuron and thus is the likely region for post-transcriptional regulation of *Ppa-cog-1*.

To investigate mechanisms for post-transcriptional regulation in the ~450 bp of the *Ppa-cog-1–3' UTR*, we looked for putative miRNA binding sites that are conserved in the putative *cog-1* 3' UTR sequences of other *Pristionchus* species (*P. exspectatus, P. arcanus, P. mayeri, P. entomophagus, P. fissidentatus*)(Fig 11A). We detected several highly conserved regions, 4 of which contain seed regions with high scores to cognate *P. pacificus* miRNA binding partners (sites A, B, C, D) that are comparable in score to the higher scoring *C. elegans lsy-6* binding site in the *Cel-cog-1* 3' UTR (162) [19]: *mir-2251b* (169), *mir-81* (159), *mir-8353* (157), *mir-8345* (149), and *mir-8364f* (151)(Fig 11B). Since functional miRNA interactions often have multiple binding sites in their target mRNA, we also looked for additional binding sites corresponding

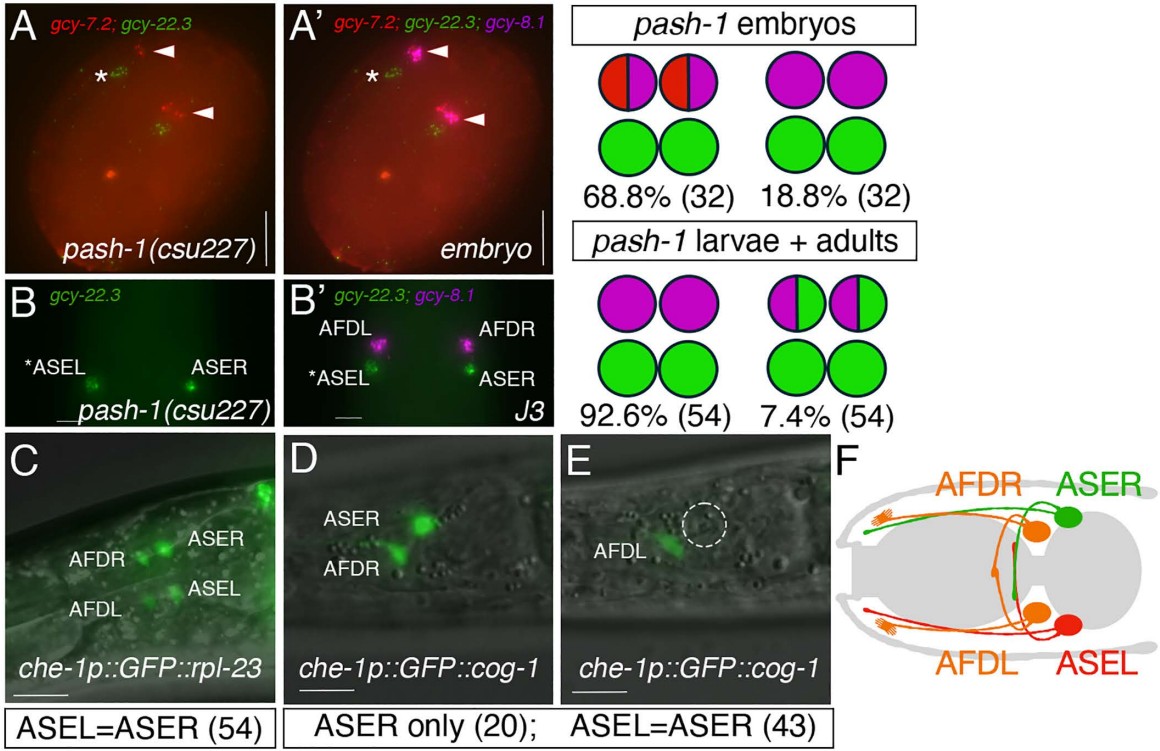

**Fig 10. miRNA biosynthesis and cog-1 3' UTR mediate the regulation of AFD and ASEL fates.** (A) HCR-FISH of a *pash-1(csu227)* late embryo shows lack of *gcy-7.2* expression in the ASEL but misexpression in the AFDs. Both ASEs express only *gcy-22.3(B2)*. (A') The sample in (A) shows co-expression of *gcy-7.2(B4)* and *gcy-8.1(B5)* in the AFDs. (B) A J3 larva shows only *gcy-22.3(B2)* expression in both ASEs. (B') *gcy-7.2(B4)* is no longer detectable and the AFD neurons show wild-type-like fate with only *gcy-8.1(B5)* expression. (C) A maximum projection of an adult *che-1p::GFP::rpl-23* 3' UTR transgenic worm showing expression in the ASEL, ASEL, and both AFDs. Equal GFP expression was detected in both ASE neurons. (D-E) An adult *che-1p::GFP::cog-1* transgenic worm with expression in the ASER but not in the ASEL. Using GFP expression in the AFDs as a control for transgene expression in mosaic animals, the *che-1p::GFP::cog-1* worms more frequently show expression only in the ASER but not in the ASEL than the *che-1p::GFP::rpl-23* worms (Fisher's Exact Test P < 0.0001). (F) A schematic diagram of the four *che-1*-expressing amphid neurons with three possible neuronal subtypes in *P. pacificus*. Scale bars represent 5 μm. See Table 5 for details.

to these candidate miRNAs. There are two possible *mir-2251b* sites with perfect 5' seed pairing 48 base pairs apart (sites A1 and A2), two possible *mir-81* sites, but only one with perfect seed pairing (site B), two possible *mir-8353* sites, also one with perfect seed pairing (site B, adjacent to the *mir-81* site), and two possible *mir-8345* sites, also with perfect seed pairing (sites C1 and C2). *Mir-8364f* has no other detectable possible binding site but is highly conserved among the 6 *Pristionchus cog-1* homologs (site D). There are also other regions with high conservation which could also play other regulatory roles.

Next, we mutated the putative miRNA binding sites to determine if the conserved *Ppa-cog-1* 3' UTR regions are required for establishing asymmetric expression of the *gcy* effector genes. We are mindful of previous findings that perfect seed pairing does not necessarily predict functional miRNA-target interaction, and that miRNA-target interactions are highly dependent on the 3' UTR context [56,57]. Therefore, to test if the conserved sites are involved in ASE and AFD patterning, we generated multiple combinations of substitutions and short indels, as well as a large 337 bp deletion encompassing all 4 of the putative miRNA binding sites. We found that various substitutions and short indels in site D alone (*csu252*), in both A sites (*csu256*), and A1 with D (*csu254*) resulted in a minority of samples with hybrid ASE/AFD fate (12–24%), predominantly with the *gcy-7.2* ASEL marker (Table 6 and Fig 11C). Furthermore, a 2-base pair

Table 5. Expression phenotypes of *Ppa-pash-1* mutants. HCR-FISH using *gcy-8.1*(B5-magenta):AFD, *gcy-7.2* (B4-red):ASEL, *gcy-22.3* (B2-green):ASER.

| | (icon 1) | (icon 2) | (icon 3) | (icon 4) | Other |
|---|---|---|---|---|---|
| Wild type (PS312) J2-Ad (51) | — | — | — | — | — |
| pash-1(csu227) J2-Ad (54) | — | 7.4% (4) | 92.6% (50) | — | — |
| pash-1(csu227) J1 only (10) | 0% | 10% (1)** | 90% (9) | — | — |
| pash-1(csu227) embryos (32) | 68.8% (22)* | — | 18.8% (6) | 9.3% (3) | 3.1% (1) |

The most frequent category is highlighted in red.

Note: The 78% of embryos with at least one ASEL expression completely lose ASEL marker expression in the post-embryonic stages.

*gcy-7.2 misexpression in either AFDL or AFDR.

**gcy-7.2 misexpression in only one AFD neuron.

substitution/deletion in one of the two *mir-2251b* binding sites (site A1, *csu239*) resulted in a much higher percentage of ectopic *gcy-7.2* expression in the AFD neurons (48%, 24 out of 50)(Table 6 and Fig 11C). We also observed in *csu239* weak expression of the ASER marker in the AFD neurons (18%, 9 out of 50), such that the AFDs express all 3 *gcy* markers. Surprisingly, when we introduced deletions to both sites A and D in *csu253,* we observed both *gcy-7.2* and *gcy-22.3* expression in the ASER, resulting in a hybrid ASEL/ASER expression (2xASEL*;~51%, 36 out of 70). Since *csu253* shares the same "TTT" deletion in site A1 as *csu254*, which only has a weak ASE/AFD hybrid phenotype, we attributed the *csu253* 2xASEL* phenotype to the 8-nt deletion of site D, the most conserved region for potential *miR-8364* binding. To test if the *csu253* 2xASEL* phenotype is due to a *cog-1* loss-of-function phenotype, we generated frameshift mutations in the *cog-1* second exon that is predicted to result in truncated COG-1 proteins (*csu257, csu258*)(S6 Fig), and found that indeed *csu253* phenocopies the *cog-1* loss-of-function phenotype (60% and 48% 2xASEL*, respectively)(Table 6 and Fig 11C). Lastly, the phenotype classes of the *Ppa-cog-1(csu255)* allele containing a large 337 bp deletion did not overlap either with those in the *csu239* or the *csu253* alleles. Specifically, in *Ppa-cog-1(csu255)* we observed *gcy-22.3* misexpression in the ASEL with or without *gcy-7.2* misexpression in the AFD neurons (2xASER*; 62%, 31 out of 50), resembling the 2xASER* *Ppa-die-1(csu225)* phenotype and mirroring the 2xASEL* *Ppa-cog-1(csu253)* phenotype (Table 6 and Fig 11C). Taken together, the *cog-1* 3' UTR contains at least 3 discrete *cis*-regulatory regions with strong miRNA binding potential mediating the repression of ASER fate in the ASEL throughout the 337 bp region (mainly due to site C, see below), ASEL fate in the ASER via binding site D, as well as the repression of ASE fate (primarily ASEL) in the AFD through binding A and D binding sites.

To validate these conserved 3' UTR regulatory sites are potential miRNA binding sites, we focused on a miRNA predicted to bind to regions C1 and C2. Although the *C. elegans lsy-6* and *miR-8345-3p* share high conservation in their seed regions [29], the *miR-8345* stem region itself shows poor complementarity while the loop region is predicted to form a tighter hairpin compared to the *lsy-6* secondary structure (Fig 12A-12B). We were therefore surprised to find that deletion alleles in the seed region of *mir-8345* exhibited a stronger 2xASER phenotype than the *cog-1(csu255)* mutant with the large deletion in the 3' UTR (Fig 12C-12G and Tables 6, 7). Compared to *cog-1(csu255),* which exhibited only partial ASEL transformation to ASER, a complete deletion of the seed region of *miR-8345(csu259)* resulted in the 2xASER phenotype in 82% of the mutants (n=51). Even a partial deletion nearby the seed region in *miR-8345(csu265)* yielded a strong 2xASER phenotype in 71% of the mutant animals (n=52). In addition to a very low hybrid ASEL/R fate, the deletions in the mature

**Fig 11. The *cog-1* 3' UTR contains multiple predicted miRNA binding motifs necessary for proper AFD and ASE specification.** (A) Sequence alignment of *cog-1* 3' UTR regions in *P. pacificus*, *P. exspectatus*, *P. arcanus, P. mayeri, P. entomophagus,* and *P. fissidentatus*. Yellow boxes highlight the 4 aligned regions containing 20-23 bp recognizable miRNA binding sites. Sequences in bold blue highlight potential *mir-2251b* 3' seed pairing region (sites A1 and A2), bold green sequences highlight potential *mir-8345* 3' seed pairing region (sites C1 and C2), and bold purple sequences highlight a potential *mir-8364f* 3' seed pairing region (site D). Bold red sequences highlight potential 3' seed pairing for *miR-81* while italic red sequences denote *mir-8353* 3' seed pairing, with overlapping seed pairing sequences in italic bold red (site B). Unanimous conserved sequences are marked by asterisks. The 2 arrowheads above delineate the deletion in *csu255* while the 2 arrows below show the positions of the two crRNAs used for generating the mutations. The phylogenetic relationship of the 6 *Pristionchus* species is shown. Sequences in green indicate complementarity between miRNA and its *cog-1* 3' UTR target. (B) The miRNA binding scores and free energy values of the 5 individual candidate binding sites are shown. (C) CRISPR/Cas9-induced mutations in the ~450 bp *Ppa-cog-1* 3' UTR and coding region. Sequences in green denote substitutions, red for deletions, and blue for insertions. HCR-FISH of 7 *cog-1* alleles probed with *gcy-8.1(B5), gcy-7.2(B4),* and *gcy-22.3(B2)*. *Ppa-cog-1(csu255)* is predicted to delete all candidate

miRNA binding sites. Images of J2-J3 stage hermaphrodites representing the most abundant mutant phenotype category for each mutant strain, indicated in red. Additional details are listed in Table 6. Putative miRNA binding complementarity is on the 3p strand, except for *mir-2a*, which is on the 5p strand. Scale bars represent 5 μm.

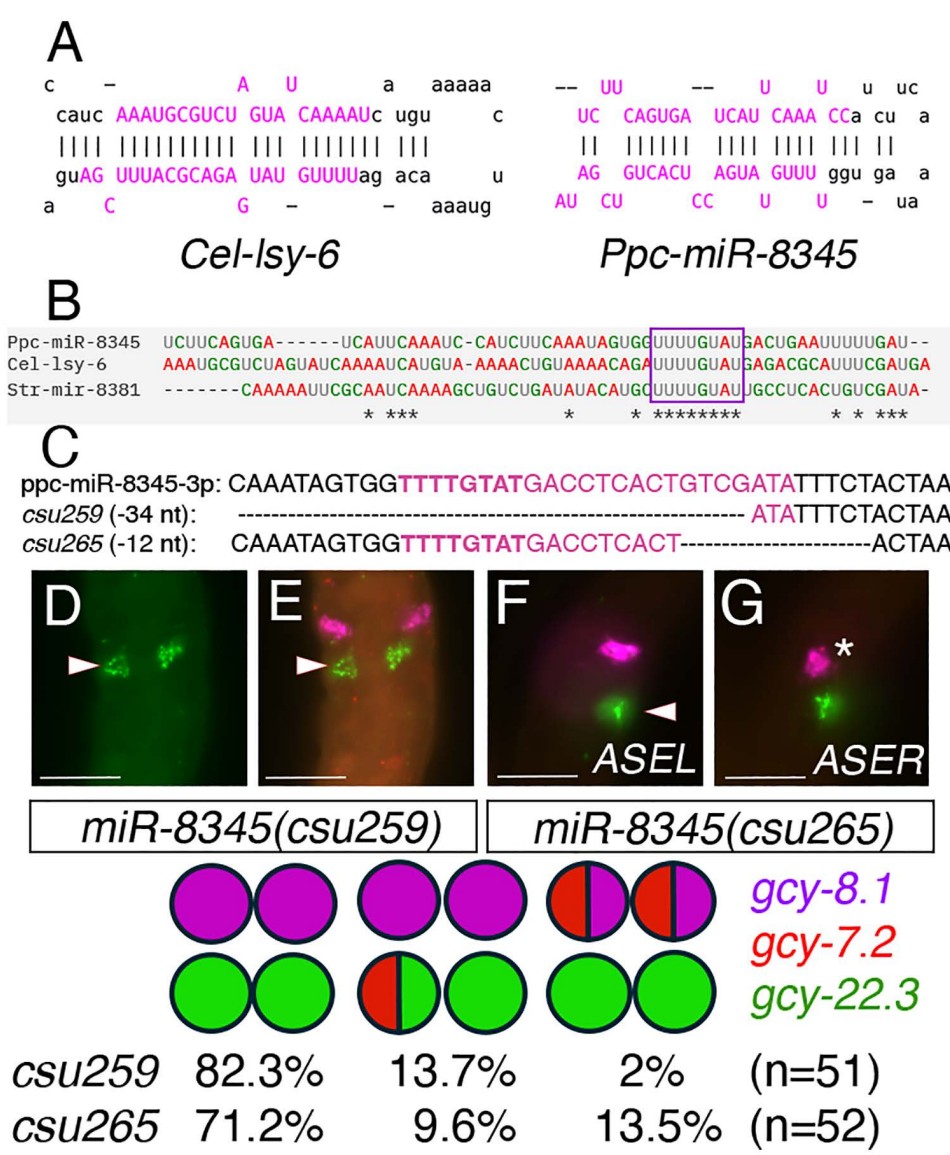

**Fig 12. Deletions in *Ppc-miR-8345* results in 2xASER phenotype.** (A) Stem-loop structures show *Ppa-mir-8345* has more mismatches in the stem region and a tighter loop than *Cel-lsy-6* (www.miRBase.org). (B) Sequence alignment of precursor miRNAs *Ppc-mir-8345*, *Cel-lsy-6* and *Str-mir-8381* (*Strongyloides ratti*), which all share the 8-nt "UUUUGUAU" seed sequence (boxed; Clustal-omega). *Denote conserved sequences. (C) DNA alignment of deletions in *miR-8345(csu259)* and *miR-8345(csu265)*. Bold letters indicate the seed sequence of *Ppc-miR-8345-3p*. (D-E) *Ppa-miR-8345(csu259)* J3 larva with ectopic *gcy-22.3* expression in the ASEL without *gcy-7.2* expression (arrowheads). (F-G) *Ppa-miR-8345(csu265)* J3 larva with ectopic *gcy-22.3* expression in the ASEL (arrowhead) as well as ectopic *gcy-7.2* expression in the AFD (*). The 2xASER phenotype is highly represented in the top 3 phenotypes of the *miR-8345* alleles.

**Table 6. Post-embryonic expression (J2-Ad) phenotypes of *Ppa-cog-1* 3' UTR and loss-of-function mutants (*csu257, csu258*). HCR-FISH using *gcy-8.1*(B5-magenta):AFD, *gcy-7.2* (B4-red):ASEL, *gcy-22.3* (B2-green):ASER.**

| | ◉◉/◉◉ (1) | ◉◉/◉◉ (2) | ◉◉/◉◉ (3) | ◉◉/◉◉ (4) | ◉◉/◉◉ (5) | ◉◉/◉◉ (6) | ◉◉/◉◉ (7) | Other |
|---|---|---|---|---|---|---|---|---|
| Wild type (PS312) (51) | 100% (51) | — | — | — | — | — | — | — |
| *die-1(csu225)* (52) | — | 73.1% (38) | 27%* (14) | — | — | — | — | — |
| *cog-1(csu239)* (50) 3' UTR | 32% (16) | — | — | — | — | 48% (24) | 18% (9) | 2% (1) |
| *cog-1(csu252)* (51) 3' UTR | 82.3% (42) | — | — | — | — | 13.7%* (7) | 2% (1) | 2% (1) |
| *cog-1(csu254)* (51) 3' UTR | 88.2% (45) | — | — | — | — | 11.8%** (6) | — | — |
| *cog-1(csu256)* (50) 3' UTR | 72% (36) | — | — | — | — | 24%* (12) | 4% (2) | — |
| *cog-1(csu253)* (70) 3' UTR | 31.4% (22) | — | — | 32.9% (23) | 18.6% (13) | 17.1% (12) | — | — |
| *cog-1(csu257)* (50) coding | 24% (12) | — | — | 26% (13) | 34% (17) | 16% (8) | — | — |
| *cog-1(csu258)* (50) coding | 42% (21) | — | — | 38% (19) | 10%* (5) | 6% (3) | 4%* (2) | — |
| *cog-1(csu255)* (50) 3' UTR | 28% (14) | 56% (28) | 6%* (3) | — | — | — | — | 10% (5) |

Wild type and *die-1* are shown for comparison. The most frequent category is highlighted in red.

*Either one or both AFDs expressing the ASEL marker *gcy-7.2*.

**Only one AFD expresses the ASEL marker *gcy-7.2*.

*miR-8345(csu259)* sequence resulted in ~86% 2xASER phenotype, in between the spectrum of *cog-1(csu255)*(62% hybrid 2xASER) and *pash-1(csu227)*(100% complete 2xASER) phenotypes (Tables 5, 6, 7 and Fig 12).

## Discussion

In *P. pacificus*, the terminal selector transcription factor *Ppa*-CHE-1 is expressed in both the chemosensory ASE neurons and the thermosensory AFD neurons [16]. In this study we confirm that *Ppa-che-1* is necessary for the expression of the ASEL (*gcy-7.1, gcy-7.2, gcy-7.3*) and ASER (*gcy-22.1, gcy-22.3, gcy-22.5*) receptor-type guanylyl cyclases, and that *Ppa*-CHE-1's role as a major positive regulator extends to the expression of AFD-type guanylyl cyclases (*gcy-8.1, gcy-8.2, gcy-8.3*). The *C. elegans* CHE-1 is known to be required to specify both the ASE neuron class as well as the ASE left-right lateral asymmetry when working together with miRNA and other transcription factors, including DIE-1 and COG-1 [28,37]. This is the first study on the genetics of neuronal asymmetry in *P. pacificus* that impact the developmental trajectories of the ASE and AFD neurons.

To investigate the regulatory architecture that supported the independent recruitment of rGCs, we initiated a limited forward genetic screen for 2xASER mutants using the ASER-specific *gcy-22.3p::gfp* reporter and identified the *Ppa-die-1* homolog as a negative regulator for ASER fate in ASEL and AFDs. Using *Ppa-die-1* as a genetic entry point, we used reverse genetics to identify other negative regulators of ASEs in the AFDs (*Ppa-ttx-1, Ppa-tax-2, Ppa-tax-4*), as well as negative regulators of ASER in the ASEL via the *Ppa-cog-1* 3' UTR. Surprisingly, the 6 major phenotype classes observed in the mutants of components of the double negative feedback loop– *die-1, cog-1,* and *ttx-1*– could be recapitulated in 3 separate mutation classes in the *Ppa-cog-1* 3' UTR. Furthermore, we have identified *miR-8345* as one of the miRNAs that likely acts through the *Ppa-cog-1* 3' UTR to determine amphid neuronal types as well as ASE-subtype lateral asymmetry.

Table 7. **Expression phenotypes of *Ppa-miR-8345* mutants**. HCR using *gcy-8.1*(B5-magenta):AFD, *gcy-7.2* (B4-red):ASEL, *gcy-22.3* (B2-green):ASER.

| | | | | |
|---|---|---|---|---|
| Wild type (PS312) J2-Ad (51) | —— | —— | —— | —— |
| *miR-8345(csu259)* J2-Ad (51) | 84.3% (43) | 2.0%* (1) | 13.7% (7) | —— |
| *miR-8345(csu265)* J2-Ad (52) | 71.2% (37) | 13.5% (7)* | 9.6% (5) | 3.9% (3)* |
| *miR-8345(csu259)* J1 (11) | 90.1% (10) | 9.1% (1)* | | |
| *miR-8345(csu259)* embryos (32) | 31.2% (10) | 62.5%* (20) | 3.1% (1) | —— |

Wild type is shown for comparison. The most frequent category is highlighted in red.

*Either one or both AFDs expressing the ASEL marker *gcy-7.2*.

### Left/right asymmetric *gcy* gene expression pattern is convergent

Based on the phylogenetic relationship among the chemosensory guanylyl cyclase receptors and their endogenous mRNA expression profiles, along with the involvement of CHE-1 in specifying AFD fate, lateral asymmetry in *P. pacificus* rGC expression likely evolved independently from the *C. elegans* lineage. Significant evolutionary divergence in chromosomal rearrangement within the *Cel-gcy-5* subfamily (*Cel-gcy-1/2/3/4/5*) of ASER-expressing *gcy* genes was already evident even between the much closer related *C. elegans* and *C. briggsae*. Specifically, *C. briggsae* does not share the adjacency of the *Cel-gcy-1, Cel-gcy-2, Cel-gcy-3* paralogs, and reporter transgene swaps between the two *Caenorhabditis* species show changes in the trans-acting factors of *Cel-gcy-4*, which lacks asymmetric expression in *C. briggsae* [21]. In *P. pacificus,* none of the left/right-expressing *gcy* genes are immediately adjacent to each other, with *Ppa-gcy-22.1* and *Ppa-gcy-22.2* being the closest to each other but separated by two intervening genes and oriented on opposite strands, suggesting they had undergone a less recent gene duplication events than the *C. elegans gcy-5* subfamily members. Whereas all of the *C. elegans* ASER-expressing *gcy* genes are located on Chromosome II (except for *Cel-gcy-22*), the *P. pacificus* ASER-expressing "*Ppa-gcy-22* subfamily" genes are spread across 3 different chromosomes, including *Ppa-gcy-22.5* on Chromosome X. Based on its unique chromosomal location and independence from *Ppa-die-1* repression, it is interesting to speculate that *Ppa-gcy-22.5* is a nascent ASER terminal identity gene. However, it remains unclear if the left/right asymmetric expression was established prior to the dispersal of the *Ppa-gcy-22* subfamily members during evolution, or separate members were subsequently acquired into the regulatory architecture to enable left/right asymmetric expression.

### *die-1* and multiple miRNAs act through *cog-1* to pattern ASE asymmetry

Whereas the 2xASER *Cel-die-1* mutant phenotype is a complete transformation of the ASEL into ASER fate, the 2xASER* phenotype of *Ppa-die-1* shows both right and left ASE marker expression in the ASEL neuron, along with a gain of ASEL marker expression in the AFDs. We infer from this result that *Ppa*-DIE-1 is necessary to repress ASER fate in the ASEL, as well as ASEL fate in the AFDs. Moreover, the repression of ASER fate in ASEL and AFDs appear to be developmentally separable since only 26% of the *Ppa-die-1* embryos compared to 100% of the *Ppa-die-1* post-embryonic stages exhibited

the 2xASER* phenotype. Thus, repression of ASER fate in the ASEL by *Ppa*-DIE-1 is primarily required to maintain proper ASEL fate after embryogenesis.

In contrast, the *Ppa-pash-1* adult phenotype shares more similarity to the *Cel-die-1* and *Cel-lsy-6* phenotypes (complete loss of ASEL fate; gain of ASER fate in ASEL) [19,25,36] compared with the *Ppa-die-1* phenotype (primarily gain of ASER in ASEL and misexpression of ASEL in AFDs), which may hint at the existence of a second effector working in parallel with Ppa-DIE-1, *i.e., Ppa*-DIE-1 does not mediate all of the repression of *Ppa-cog-1* in the ASEL. Given the absence of the 2xASER* phenotype in *Ppa-die-1* mutants when *gcy-22.5* instead of *gcy-22.3* was used as the ASER marker, there may be paralog-specific regulation by DIE-1-dependent along with DIE-1-independent processes. Alternatively, it is also possible that the negative regulation of *gcy-22.5* by *Ppa-die-1* in the ASEL requires a more severe *Ppa-die-1* allele than for *gcy-22.3*.

Despite a complete deletion of all putative miRNA binding sites in the 3' UTR of *Ppa-cog-1(csu255)*, which resulted in the 2xASER* phenotype, we still observed almost wild-type expression of the ASEL marker, *gcy-7.2* (only 6% misexpress *gcy-7.2* in the AFDs, Table 6). It is possible that the *Ppa-die-1(csu225)* mutation selectively affected only one of the two predicted protein isoforms responsible for post-transcriptional repression of ASER fate without affecting the other protein isoform promoting ASEL fate. Alternatively, given the 2xASER phenotype of *Ppa-pash-1(csu227)*, the hybrid ASEL/ASER phenotype in the ASEL neuron suggests that *Ppa*-PASH-1 may control several miRNAs regulating the ASE asymmetry, including both the *miR-8345* regulating *Ppa-cog-1* as well as another yet-to-be identified "X" regulating a repressor of *gcy-7.2* in the ASEL neuron (Fig 13). However, we are cautious with this interpretation since both *Ppa-die-1* and *Ppa-pash-1* alleles are not functional nulls (due to embryonic lethality), and the exact effects of the *Ppa-pash-1* mutation on the maturation of miRNAs acting on *cog-1* is unknown, so the impacts of their mutations on their individual targets may be disproportionate. Thus, the reduction-of-function *Ppa-pash-1* allele revealed multiple possible miRNAs function to repress ASER fate in the ASEL neuron, as well as to repress ASE fates in the AFD neurons.

The mutant phenotypes of *Ppa-cog-1(csu239)* with point mutations in site A1 and *Ppa-cog-1(csu253)* with an 8-nt deletion in site D do not overlap with the *Ppa-cog-1(csu255)* large deletion phenotype that includes all of the putative miRNA binding sites. This result is unexpected because: (1) alleles containing other types of mutations in site A1 and a second site resulted in only the ASE/AFD hybrid phenotype (*csu254* and *csu256*), suggesting that either only the specific *csu253* deletion in site A1 can alter ASEL fate, or the effect of the deletions in both sites A1 and D is synergistic; (2) The large deletion in *csu255* removed all of the predicted miRNA seed regions and thus could have resulted in the recapitulation of the *csu239* hybrid ASEL/AFD as well as the *csu253* 2xASEL* phenotypes. Instead, the *csu255* deletion resulted in a gain-of-function 2xASER phenotype similar to the *Cel-cog-1(ot123)* allele with a deletion in the major *lsy-6* binding site in the 3' UTR [36,58], which suggests the intervening miRNA binding sites (B and C) are important for preventing *Ppa-cog-1* expression in the ASEL. Among the 4 possible distinct miRNA binding sites, the C1 and C2 binding sites are the most conserved sequences with perfect complementarity to the seed sequences of *lsy-6* and *ppc-miR-8345-3p* (Fig 12) [29]. The likelihood that this *miR-8345/lsy-6* locus acts through the *cog-1* 3' UTR is further bolstered by the near complete penetrance of the 2xASER phenotype in the two *miR-8345* mutants (*csu259* and *csu265*). Hence, *csu255* likely led to a gain-of-function misexpression of *Ppa-cog-1* in the ASEL, although the *Ppa-cog-1* 3' UTR may not be the only target of PASH-1-dependent miRNAs since neither the *csu255* large deletion nor the *miR-8345* mutations resulted in the 100% conversion of ASEL into ASER phenotype observed in the *Ppa-pash-1(csu227)* mutant. It is also possible that sequences outside of the deletion region in the rest of the ~450 bp 3' UTR may contain additional unrecognized miRNA binding sites or other sequence context necessary for both negative and positive interactions among the various post-transcriptional regulators acting through this *Ppa-cog-1* 3' UTR regulatory nexus; [3] Lastly, the *csu253* 2xASEL* phenotype resembled *cog-1* reduction-of-function mutants, and as such site D may serve as a positive post-transcriptional regulatory element rather than a site mediating repression by a miRNA.

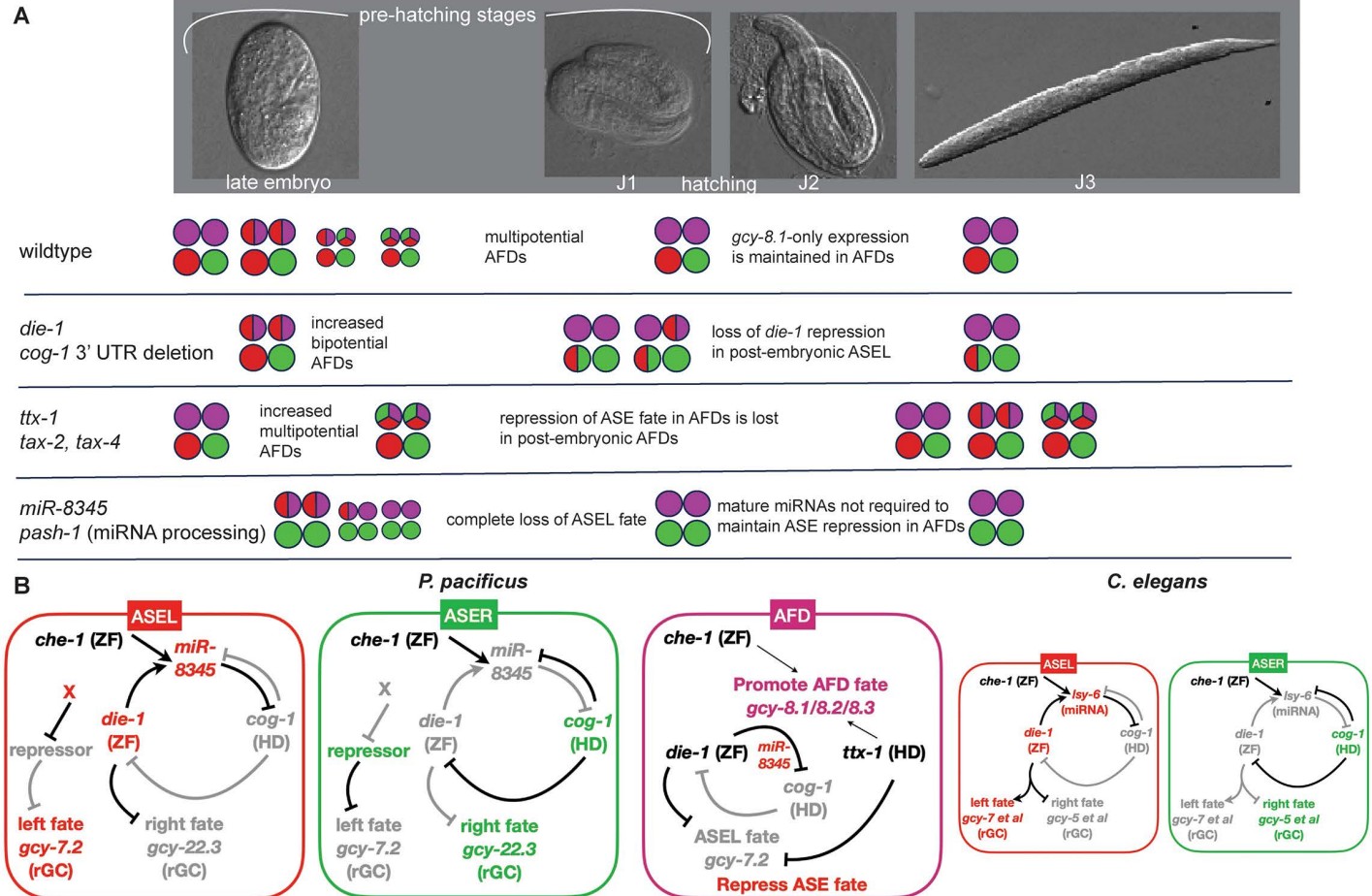

**Fig 13. Summary and molecular regulatory model.** (A) Left-right ASE asymmetry is specified by late embryogenesis while the restriction of ASE fates in AFD occurs shortly before the end of embryogenesis. (B) A molecular regulatory model for neuronal patterning of AFD and ASE fates in *P. pacificus*, compared to *C. elegans*. Black line denotes the interaction is active while gray line denotes the interaction is silent. ASE asymmetry based on the double-negative interactions is likely to be conserved with *C. elegans*, except for a predicted factor "X" regulating the expression of *gcy-7.2* in the ASEL. *che-1* is required for all ASE-subtype *gcy* expression, and *Ppa-miR-8345/lsy-6* represses *cog-1* in the ASEL. Expression of *gcy-8* paralogs in the AFDs are mediated partially by *che-1* and *ttx-1*. *die-1*, *ttx-1*, and *Ppa-miR-8345* are involved in the repression of ASE-subtype fates.

## Restriction of ASE fate in AFD neurons

We have revealed the conserved regulators for ASE asymmetry to also function to repress ASE fates in the AFD neurons. Both the reduction-of-function *Ppa-die-1* mutant and every mutation we tested in the *Ppa-cog-1* 3' UTR resulted in some degree of misexpression of ASEL fate in the AFDs. If *Ppa*-DIE-1 function in the ASEL is conserved in its activation of *miR-8345* to repress *Ppa-cog-1* post-transcriptionally through its 3' UTR, then it is likely that *Ppa*-DIE-1 in the AFDs also similarly acts upstream to repress *Ppa-cog-1*. Whereas *Ppa*-DIE-1 represses ASER fate (*Ppa-gcy-22.3*) in the ASEL, we infer from the results that *Ppa*-DIE-1 may also repress ASEL fate (*Ppa-gcy-7.2*) in the AFDs. In support of the separation of ASE versus AFD fate regulation, the *Ppa-die-1(csu225)* and the *Ppa-cog-1(csu255)* 3' UTR deletion mutants share the 2xASER* phenotype, while *Ppa-ttx-1(csu150)* and the *cog-1(csu239)* 3' UTR site A1 mutants both show significant misexpression of ASE markers in the AFD neurons. Although *Ppa-die-1(csu225)* is a hypomorphic allele due to the single amino acid change in one of the two Zn-finger domains, its incomplete and hybrid 2xASER* phenotype may not necessarily be

on the account of *Ppa-die-1(csu225)* being a weak allele, since a similar allele with a missense mutation in the Zn-finger domain, *Cel-die-1(ot100)*, has a 100% penetrance of the 2xASER phenotype [36]. Null *die-1* alleles in *C. elegans* are embryonic lethal while viable *Cel-die-1* mutations with the 2x ASER phenotype cluster at the C-terminus [36,59]. Thus, if the *die-1* mutants between the two species are of comparable severity, then the difference in the *die-1* phenotypes may be due to different underlying regulatory roles between these orthologs. To accommodate the terminal selector role of *Ppa*-CHE-1 in both ASE and AFD neurons in *P. pacificus*, the *Ppa-cog-1* 3' UTR acts as a *die-1*-dependent "toggle" in the specification of ASEL and AFD.

We have revealed several other negative regulators of ASE terminal effector genes acting in the AFD neurons– *Ppa-ttx-1, Ppa-tax-2,* and *Ppa-tax-4*. While the role for the CNG channels seems superficially identical to their counterpart in *C. elegans* (restricting ASE fate in the AFDs) [36], the degree of *Ppa*-TTX-1 involvement in restricting AFD potential is more extensive. The hypomorphic *Ppa-ttx-1* allele revealed several key differences to the *Cel-ttx-1(p767)* reference allele [48]: (1) *Ppa-ttx-1(csu150)* are athermophilic rather than cryophilic, although this difference could be explained by linear versus radial thermotaxis assays. This defect in thermotaxis behavior is not accompanied by a gross morphological defect in the AFD neurons we can detect by light microscopy; (2) *Ppa-ttx-1(csu150)* show a temperature-dependent dauer constitutive phenotype, which could be due to defects in AFD function that results in the mutants interpreting a higher temperature than wildtype. (3) Whereas the *Cel-gcy-8::GFP* reporter has weaker expression in *Cel-ttx-1(p767)* than in wild-type adults, *Cel-gcy-8* expression is unaffected in the L1 stage. AFD-specific *Ppa-gcy-8.1* expression showed no detectable change in *Ppa-ttx-1(csu150)* mutants but only when *che-1* function was additionally compromised; (4) Whereas *Cel-ttx-1(p767)* mutants express partial AWC-like characteristics (*str-2* and *odr-3* expression)[48], ~32% of the *P. pacificus ttx-1* mutant adult animals express ASEL and ASER-specific *gcy* markers in AFD neurons (*gcy-7.2* and *gcy-22.3*, respectively). The *Ppa*-TTX-1 repression of ASE fate is likely critical in late embryogenesis, when the wild-type AFD precursors still share a common ASE chemosensory *gcy* developmental fate. In *C. elegans*, only 2% of the *Cel-ttx-1(p767)* mutants and 23% of heat-shock induced CHE-1 transgenic worms exhibited ectopic ASER-type (*gcy-5*) expression. Removal of both *Cel-ttx-1* and its co-factor for AFD specification, *Cel-ceh-14*, resulted in only ~34% of ectopic ASE marker expression in the AFD neurons [60]. Comprehensive analysis of all homeobox genes in *C. elegans* indicates the ASE, AWC, and AFD neurons share the most similar homeobox gene expression profile [61]. Therefore, it is likely that a similar regulatory architecture underlies *P. pacificus* neuronal patterning, but AWC-specific markers as well as additional homeobox genes will need to be identified to better delineate the extent of regulatory conservation.

In spite of the deep divergence between *C. elegans* and *P. pacificus*, we found the functional components of the bistable negative feedback loop consisting of *Ppa-DIE-1, Ppa-COG-1,* and *Ppa-miR-8345/lsy-6* are largely conserved, as based on their loss-of-function phenotypes that alter lateral fate in the ASE neurons. More intriguingly, the *Ppa-cog-1* 3' UTR has been exapted from an ASE laterality regulator to include negative regulation of ASE fate in AFD specification. Thus the expanded role of Ppa-CHE-1 as a terminal selector for both ASE and AFD neurons could be a case of developmental systems drift [62,63] that co-opted the *Ppa-cog-1* 3' UTR to act as a regulatory switch for the 3 neuronal fates (ASER, ASEL, AFD) along with the recruitment of five *gcy-22* paralogs for ASER-specific expression. The remaining functional binding motifs in the *Ppa-cog-1* 3' UTR could provide further opportunities to better understand the evolution of miRNAs and their target acquisitions.

## Materials and methods

### Nematode strains

*P. pacificus* and other nematode strains were maintained at ~20°C on NGM plates seeded with *E. coli* OP50 for food as described previously [64]; these are derived from standard *C. elegans* culture methods [65]. Nematode mutant strains are listed in S1 Table.

## Transgenic strains

Reporter strains were generated as reported previously [16,66]. In brief, PCR products of putative promoter regions upstream of *gcy* transcripts based on El Paco V3 gene prediction (www.pristionchus.org) were amplified by high fidelity DNA polymerase and fused to the codon-optimized GFP coding region (*pZH008* or *pSNP36* with Hygromycin$^R$) [55,67] using Gibson Assembly (New England Biolabs, MA). Following plasmid DNA purification (GeneJET Plasmid Miniprep, ThermoFisher Scientific, CA) and sequence confirmation, the reporter plasmids (2 ng/µl) and *Ppa-egl-20p::rfp,* along with PS312 genomic DNA (80 ng/µl) were individually digested with HindIII and PvuI to create the injection mixes to generate the independent reporter strains in PS312: *csuEx74[Ppa-gcy-22.1p::GFP], csuEx104[Ppa-gcy-22.2p::GFP]; csuEx102[Ppa-gcy-22.4p::GFP]; csuEx105[Ppa-gcy-22.5p::GFP]; csuEx84[Ppa-gcy-5p::GFP], csuEx108[Ppa-gcy-7.2p::GFP], csuEx101[Ppa-gcy-8.1p::GFP], csuEx100[Ppa-gcy-8.2p::GFP].* We noticed that the brighter co-injection marker with the codon-optimized RFP (*Ppa-egl-20p::opt-RFP*) shows occasional expression in the tail and the CAN neurons (lateral midbody) as well as in the ASER, as confirmed separately by co-localization with HCR using the *gcy-22.3* probe and the lack of co-localization with the *gcy-7.2* probe (n = 10). Worms expressing *Ppa-egl-20p::rfp* in the hermaphrodite tail were selected for expression analysis. We estimate that the generation of transgenic strains in *P. pacificus* was ~ 100 fold less efficient than in *C. elegans*. On average, only half of our transgene constructs yielded a stably transmitted line and of these, we obtained only one stably transmitting line per 141 injected *P. pacificus* worms.

To construct the *Ppa-che-1pei::gfp:cog-1–3' UTR* sensor strain, we swapped out the *Ppa-rpl-28* 3' UTR from *Ppa-che-1p::GFP:rpl-28 (pMM6)* with a 672 bp fragment downstream of the stop codon [16]. This putative *Ppa-cog-1* 3' UTR region was then inserted behind the translational stop of the codon-optimized GFP sequence by Gibson Assembly (New England Biolabs, MA). Reporter plasmids and primers are listed in S2 and S3 Tables.

Worms were imaged with a Leica DM6000 upright fluorescence microscope using a 40x oil objective and a Leica K5 cMOS camera. To obtain maximum projection images, we also used a Zeiss Axio Imager with Apotome 3.0 with Zen software. For left-right determination, we looked to see if the nose of the animal (after the J3 stage when the girth is wide enough) is in focus or out-of-focus when the reporter expression is in focus, as well as at the ventral orientation using the vulva or the gonad.

## Mutagenesis for *das* mutants

EMS mutagenesis was performed on *csuEx90* containing the extrachromosomal array of the *gcy-22.3p::gfp* reporter for the ASER neuron. 420 $F_1$ progeny from ~50 $P_0$ animals (840 haploid genomes) were singled onto 200 µl of OP50 and screened for $F_2$ mutants with 2 or more GFP-expressing neurons. We obtained *das-1(csu222)* and *das-2(csu223)*. *das-1(csu222)* was subsequently outcrossed twice to wild-type PS312 before characterization. *das-1(csu222)* is a recessive mutation since the cross between *das-1(csu222)* and wildtype yielded only heterozygote F1 male progeny with the wild-type 1 ASER phenotype. Whole genome sequencing by BGI America (San Jose, CA) was performed on the outcrossed *das-1(csu222)* and the unmutagenized *csuEx90* control using the DNA Nano Ball (DNBSEQ) platform with pair-end read of 150 nt at phred 33. SOAPnuke tool was used to trim adaptor and low-quality reads. Approximately 40 million clean paired reads were mapped onto the *Pristionchus pacificus* genome (El paco V3; www.pristionchus.org). From the mapped reads, we identified SNPs in coding regions that resulted in non-synonymous changes present in *csu222* but absent in *csuEx90*.

## CRISPR/Cas-9 targeted mutagenesis

CRISPR/Cas9 mutagenesis was used to generate gene-specific mutations [55,68]. crRNA and primer sequences, and induced mutations, are included in S3 Table. Specifically, Homology-Directed Repair (HDR) was used to generate targeted mutations in *die-1, cog-1,* and *pash-1*. We did not detect a temperature-sensitive Dumpy (Dpy), gonad migration (Mig),

or egg-laying (Egl) phenotype in *Ppa-pash-1(csu227)* (S7 Fig). Non-Homologous End Joining (NHEJ) was employed to target random mutations the second exon of *ttx-1* [55,68]. We used 2 crRNAs and a repair template to generate *cog-1(csu256)* with substitutions as well as *cog-1(csu255)* with a 337 bp deletion in the 3' UTR. We used a single crRNA to generate other *cog-1* alleles with 3' UTR mutations in putative miRNA binding sites as well as in the coding region (Exon 2). Unlike *C. elegans cog-1,* which has two transcripts with different starting exons (Palmer 2002), the *Ppa-cog-1* has 2 transcripts with different number of exons in the 3' end (10 and 12 exons; Trinity Transcriptome, [www.pristionchus.org](http://www.pristionchus.org)). These putative *Ppa-cog-1* null alleles show the Egl phenotype similar to the *C. elegans cog-1* alleles. Target crRNA, tracrRNA, and Cas9 nuclease were purchased from IDT Technologies (San Diego, CA). crRNA and tracrRNA were hydrated to 100 µM with IDT Duplex Buffer, and equal volumes of each (0.61 µl) were combined and incubated at 95°C for 5 minutes, then 25°C for 5 minutes. Cas9 protein (0.5 µl of 10 µg/µl) was added, then the mix was incubated at 37°C for 10 minutes. *Ppa-egl-20p::optRFP* was used as a co-injection marker. To reach a final total volume of 40 µl, the Cas9-crRNA-tracrRNA complex was combined with *pZH009* DNA (*Ppa-egl-20p::rfp*) to reach a 50 ng/µl final concentration using nuclease-free water. $F_1$ progeny were screened for the presence of *Ppa-egl-20p::RFP* expression in the tail and PCR products from candidate $F_1$'s were sequenced to identify heterozygote carriers for subsequent analysis [68]. The precise molecular lesions are shown in S6 Fig.

## Thermotaxis and chemotaxis assays

Thermotaxis assays were performed and analyzed as previously described, using a large-format linear thermal stage [69]. Worms cultured at 20°C were washed in M9 buffer and manually placed using a micropipette along the starting temperature (25°C), and allowed to travel for 60 min on a 22 x 22 cm square agar plate spanning a temperature gradient of ~21°C to ~34°C. Multi-stack images captured during thermotaxis assays were analyzed in FIJI. Images were processed to improve visibility of worms for tracking. Using the Cell Counter plug-in, each worm on the assay plate was assigned an ID number, and a subset of 10 worms per condition were randomly selected for analysis. Using the Manual Tracking plug-in in conjunction with the Cell Counter, worms were manually tracked one at a time in every frame (excluding the first 10 minutes for acclimation) for the duration of the recording or until the worm exited the field of view. FIJI tracking results were transferred to a Microsoft Excel file and analyzed using custom MATLAB scripts (WormTracker3000; [https://github.com/astrasb/WormTracker3000](https://github.com/astrasb/WormTracker3000)) [69] in order to generate plots showing multiple worm tracks associated with distance (cm) and temperature (°C) coordinates. Additionally, for each condition, the tracks for 2 "scouts" were measured to determine the maximum range of travel: 2 individuals from the assay population that traveled the farthest along either direction ("hot" versus "cold") of the temperature gradient. Results were based on 3 separate runs with both wildtype PS312 and *Ppa-ttx-1(csu150)* animals.

The chemotaxis assay for assessing response to salt gradients was adapted from *C. elegans* and *P. pacificus* chemotaxis assays as previously described [11,21,70]. In brief, overnight salt gradients were established on 10 cm chemotaxis plates containing 20 ml agar (5 mM KPO4, 1 M $CaCl_2$, 3% Bacto-agar, 1 mM $MgSO_4$) by adding 10 µl of 2.5 M salt solutions for 16 hours. Alternatively, agar containing 25 mM ($NH_4Cl$, $NH_4I$) or 50 mM (NaCl) were used to test for responses to individual salt ions. Following the establishment of the overnight point gradient, another 4 µl of the same salt solution or water control was added to reinforce the gradient 4 hours before the assay. Just prior to the assay, 1 µl of 1 M sodium azide was added to both the attractive salt (A) and the control (C) spots. *P. pacificus* J4 to adult hermaphrodites from near-saturated cultures were washed 3x with distilled water and collected by centrifuging at 2000 rpm for 2 minutes. Approximately 200 worms were loaded onto the edge of each assay plate between the gradient sources, and at least 10 combined worms have to reach the scoring arenas to be included in the analysis. Multiple trials over several days totaling at least 10 assays were conducted and averaged for each condition. The Chemotaxis Index (CI) for each end-point assay plate is defined as (A - C)/(A + C).

## Dauer formation

Embryos were synchronized by a 4-hour egg laying on OP50 and cultured at 20°C, 25°C, and 27°C. J3 and Daf-c dauer larvae were subsequently scored on day 4 on well-fed plates.

## mRNA *in situ* hybridization chain reaction (HCR-FISH)

We employed the third-generation HCR technology with split-initiator probe pairs (v3.0) and three differently conjugated fluorophores (B2, B4, B5; see S4 Table)(Molecular Instruments, CA) with 30 probe pairs covering all genes except for *Ppa-die-1*. We ordered probe sets as DNA oligo pools "oPool" at 50 pmol (IDT, San Diego, CA) and used a 50x higher concentration of probes than previously published (100 pmol) [41]. For certain samples, we also took Z-stacks using the Leica DM6000 followed by 3D rotation in Fiji to resolve staining bodies in close proximity to each other. We wrote a Python script to standardize the tabulation and visualization of the of expression phenotypes as "pies" (https://github.com/honglabcsun/Neuronal_Asymmetry). We examined all post-embryonic stages after hatching that includes J2 to adult stages, as well as the pre-hatch J1 stage, which may also include J2 larvae after the J1 molt since the the J1 cuticle (exuvia) is not visible in fixed samples. Embryonic samples that express the gcy genes are predominantly late embryonic stages (3-fold) that are readily distinguished from the pre-hatch J1 by their tighter packing inside the eggshell.

## miRNA analysis

To identify phylogenetic footprints of conserved regulatory regions in the 3' UTR of *cog-1* homologs, we obtained non-coding sequences from approximately 1000 bp after the stop codon from *P. exspectatus, P. arcanus, P. entomophagus, P. mayeri,* and *P. fissidentatus* from whole chromosome assemblies (www.pristionchus.org) [71,72]. All *cog-1* homologs reside on the syntenic Chromosome II. DNA alignment was performed using Clustal Omega (ebi.ac.uk) [73] followed by manual gap arrangement for sites C1 and C2 to identify conserved regions containing 6- to 8-basepair stretches that returned with high probability scores for miRNA binding to previously identified miRNA in the *P. pacificus* genome from miRBase (www.mirbase.org) and using RegenDbase for miRNA target detection with strict 5' seed-pairing and a threshold score of 140 (regenbase.org/tools/miranda) [74–76]. For miRNA genes, the same prefix 'Ppa-' is used as the nomenclature for protein-coding genes (*e.g., Ppa-miR-2251b-3p*), whereas the corresponding mature miRNA uses the prefix '*ppc-*' in accordance with miRbase (S8 Fig).

To characterize the miRNA processing defects of the *Ppa-pash-1(csu227)* mutant, we performed next-generation small RNA sequencing using 5 µg of phenol-extracted and ethanol precipitated total RNA from mix-stage cultures (RIN > 9). The libraries were prepared with QIAseq Library Preparation Kit (Qiagen, Germany) for sequencing on the MGI 400 sequencer for 150 bp pair-end sequencing. 41,435,445 and 36,088,237 reads (~90% of all reads) passed filter for wildtype and *pash-1* samples, respectively (IDseq, Davis, CA). Bioinformatic analysis for differential expression was performed using the nf-core/smrnaseq workflow (v2.3.1) and the reference miRNA datasets for *P. pacificus* (GSM1016459, GSM1016460, GSM1016461)[29]. The raw pair-end reads for the small RNAseq have been submitted in the Sequence Read Archive (SRA) as Accession PRJNA1413424.

## Phylogenetic tree

The amino acid sequences of potential homologs were first identified by BLASTX searches on WormBase. The phylogeny trees were built using the full-length amino acid sequences in the following manner: alignment and removal of positions with gap with MUSCLE, Bayesian inference (MrBayes) and tree rendering by TreeDyn (www.phylogeny.fr) [77]. Markov Chain Monte Carlo was executed for 100,000 generations. Branch support >0.5 is shown. In cases of 1-many homology of *gcy* genes, we designated PPA13334 as the *P. pacificus gcy-5* homolog based on its conserved chromosomal location

with the (Chr. II), compared with the *gcy-5* paralog PPA02210 on Chromosome IV. In this study, we assigned the *gcy-8* paralogs differently from WormBase, which had assigned *PPA05923* as *gcy-8.2* and *PPA41407* as *gcy-8.3*.

**Nomenclature**

When unambiguous, *P. pacificus* genes are referred to without the *Ppa-* prefix; if necessary for comparison to another species such as *C. elegans (Cel-)*, the *Ppa-* prefix was then used.

**Supporting information**

**S1 Table. Nematode strains.**
(DOCX)

**S2 Table. Plasmids.**
(DOCX)

**S3 Table. Primer sequences.**
(DOCX)

**S4 Table. in situ HCR probes.**
(DOCX)

**S1 Fig. Co-expression of *gcy-22* transcripts in ASER neurons.** (A-C) *gcy-22.2* co-localized with *gcy-22.5* in a J4 hermaphrodite. (D-F) *gcy-22.4* co-localized with *gcy-22.5* in an adult hermaphrodite. (G-I) *gcy-22.1* co-localized with *gcy-22.3* in a J4 hermaphrodite. Triangles indicate co-expression of the *gcy-22* paralogs. The scale bar represents 25 µm.
(TIF)

**S2 Fig. gcy-5 transcripts co-localize with che-1p::GFP in an ASE neuron.** (A-C) A J3 hermaphrodite shows *gcy-5* co-localization with the *che-1p::GFP* reporter in an ASE neuron based on cell body position (AFD would be more anterior). The scale bar in (C) represents 25 µm.
(TIF)

**S3 Fig. die-1 is expressed in many cell types but excluded in the ASER.** (A-A') HCR FISH of a J3 larva shows *die-1(B5)* can co-express with *gcy-7.2(B4)* transcripts in the ASEL neurons. (B-B') In contrast in the same animal, *die-1(B5)* do not co-express with *gcy-22.3(B2)* transcripts in the ASER neurons. Scale bar represents 5 µm.
(TIF)

**S4 Fig. das-2 mutants show ectopic expression of ASEL and ASER gcy transcripts in AFD neurons.** (A-B) Ectopic expression of the ASER *gcy-22.3p::GFP* reporter in AFD neurons in a J3 hermaphrodite. The scale bars represent 50 µm. (C-F) Ectopic expression of the *gcy-7.2* and *gcy-22.3* transcripts were found in AFD* neurons that also co-express the *gcy-8.1* marker. "?" in (D) denotes an unidentified neuron posterior to the ASEs observed in 24% animals (12 out of 50). (F) Overlay image of all three channels (C-E). Triangles in (C) and (D) indicate neurons with ectopic expression of ASE markers in the AFD neurons. The scale bar in (D) represents 25 µm in panels C-F.
(TIF)

**S5 Fig. Changes in miRNA profile in *Ppa-pash-1(csu227)* mutants.** (A) ~90% of the RNA sequence reads for both samples passed quality filters. (B-C) Compared to wildtype, the *Ppa-pash-1* mutant has a lower percentage of reference miRNA and more miRNA variants. (D) The peak read length of mature miRNAs is 21 bp for *Ppa-pash-1* mutants (black) compared to 22 bp for wildtype (blue).
(AI)

**S6 Fig. Molecular lesions.** DNA alignment of the CRISPR/Cas9-generated *P. pacificus die-1, pash-1, ttx-1,* and *miR-8345* mutant alleles and their predicted amino acid changes. The GCT>A̲CT (A359T) substitution of *die-1(csu222, csu225)* is predicted to affect the sixth zinc-finger domain. The TC̲A>TAC̲ (S494Y) mutation of *Ppa-pash-1(csu227)* is likely to affect an RNA-binding domain, dRBD2. The in-frame 9 nucleotide complex insertion in *Ppa-ttx-1(csu150)* results in a "CVIW" replacement of the V residue near the N-terminus of the protein. Bold black letters indicate mutations. For *miR-8345* alleles *csu259* and *csu265*, magenta highlights the mature miRNA and bold sequences indicate the seed region.
(DOCX)

**S7 Fig. Morphological phenotypes of *Ppa-pash-1(csu227)* lack temperature sensitivity at 15°C and 20°C.** (A) *Ppa-pash-1(csu227)* exhibits the Dumpy (Dpy) phenotype at both temperatures. Multiple Mann-Whitney test **$P < 0.01$. (B) *Ppa-pash-1(csu227)* exhibits a gonad migration (Mig) phenotype at both tempertures. ANOVA with Sidak's multiple Comparisons test. ****$P < 0.0001$ (C) *Ppa-pash-1(csu227)* does not show an egg-holding (Egl) phenotype compared to wild type PS312. Multiple Mann-Whitney test *$P < 0.05$. Sample sizes: PS312 at 15°C (6 assays, n=274); *pash-1* at 15°C (7 assays, n=631); PS312 at 20°C (5 assays, n=461); *pash-1* at 20°C (6 assays, n=438).
(AI)

**S8 Fig. Mature *P. pacificus* miRNAs with complementary binding site sequences in the *Ppa-cog-1* 3' UTR.**
(DOCX)

**S1 Appendix. MultiQC report (html).** Differential Expression list of hairpin miRNA with volcano plot (csv and png). Differential Expression list of mature miRNA with volcano plot (csv and png). Mirtrace length plot (png). Mirtrace phred plot (png). Mirtrace complexity plot (png). Mirtrace contamination plot.
(ZIP)

## Acknowledgments

We thank the Hallem Lab for assistance with thermotaxis assays, and L. Castro, V. Le and S. Kuznyetsova for technical assistance.

## Author contributions

**Conceptualization:** Ray L. Hong.

**Data curation:** Dylan L. Castro, Ivan M. Dimov, Marisa Mackie, Heather R. Carstensen, Mary T. Barsegyan, Ray L. Hong.

**Formal analysis:** Ivan M. Dimov, Marisa Mackie, Ray L. Hong.

**Funding acquisition:** Ray L. Hong.

**Investigation:** Dylan L. Castro, Ray L. Hong.

**Methodology:** Dylan L. Castro, Marisa Mackie, Heather R. Carstensen, Ray L. Hong.

**Project administration:** Ray L. Hong.

**Resources:** Ray L. Hong.

**Software:** Dylan L. Castro.

**Supervision:** Ray L. Hong.

**Visualization:** Ray L. Hong.

**Writing – original draft:** Ray L. Hong.

**Writing – review & editing:** Dylan L. Castro, Ivan M. Dimov, Marisa Mackie, Heather R. Carstensen, Ray L. Hong.

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
