## [Decision Letter · Decision Letter 0]

29 Jul 2025

PGENETICS-D-25-00733

The rewiring of a terminal selector regulatory cascade generates convergent neuronal laterality

PLOS Genetics

Dear Dr. Hong,

Thank you for submitting your manuscript to PLOS Genetics. After careful consideration, we feel that it has merit but does not fully meet PLOS Genetics's publication criteria as it currently stands. Therefore, we invite you to submit a revised version of the manuscript that addresses the points raised during the review process.

Please submit your revised manuscript within 60 days Sep 27 2025 11:59PM. If you will need more time than this to complete your revisions, please reply to this message or contact the journal office at plosgenetics@plos.org. Please include the following items when submitting your revised manuscript:

We look forward to receiving your revised manuscript.

Kind regards,

Nathalie Pujol

Academic Editor

PLOS Genetics

Monica Colaiácovo

Section Editor

PLOS Genetics

Aimée Dudley

Editor-in-Chief

PLOS Genetics

Anne Goriely

Editor-in-Chief

PLOS Genetics

**Journal Requirements:**

https://journals.plos.org/plosgenetics/s/submission-guidelines#loc-parts-of-a-submission

4) We notice that your supplementary Figures, and Tables are included in the manuscript file. Please remove them and upload them with the file type 'Supporting Information'. Please ensure that each Supporting Information file has a legend listed in the manuscript after the references list.

Potential Copyright Issues:

i) Please confirm (a) that you are the photographer of 2, and 10C, or (b) provide written permission from the photographer to publish the photo(s) under our CC BY 4.0 license.

6) When completing the data availability statement of the submission form, you indicated that you will make your data available on acceptance. We strongly recommend all authors decide on a data sharing plan before acceptance, as the process can be lengthy and hold up publication timelines. Please note that, though access restrictions are acceptable now, your entire data will need to be made freely accessible if your manuscript is accepted for publication. This policy applies to all data except where public deposition would breach compliance with the protocol approved by your research ethics board. If you are unable to adhere to our open data policy, please kindly revise your statement to explain your reasoning and we will seek the editor's input on an exemption. Please be assured that, once you have provided your new statement, the assessment of your exemption will not hold up the peer review process.

7) Please amend your detailed Financial Disclosure statement. This is published with the article. It must therefore be completed in full sentences and contain the exact wording you wish to be published.

2) If any authors received a salary from any of your funders, please state which authors and which funders..

8) Kindly revise your competing statement to align with the journal's style guidelines: 'The authors declare that there are no competing interests.'

**Reviewers' comments:**

Reviewer's Responses to Questions

**Comments to the Authors:**

Reviewer #1: In this manuscript the authors use the comparison between C. elegans and P. pacificus as a way to explore how gene regulatory evolution can be linked to phenotype, specifically in the development of the nervous system. The authors make a number of findings about how gene expression regulation is implicated in the development of asymmetry in the P. pacificus nervous system. The work is clear and most of the experiments are performed well. I think one would have to say that the conclusions are of specialised interest and from only one species, it is difficult to make generalisations about how regulatory systems evolve. However, detailed exploration of neuronal asymmetry in emerging model organisms will be important in moving beyond studies in a few widely separated systems (ie C. elegans, Drosophila, mouse etc). Collecting data from different systems is clearly the first, necessary, step in this process so the present work is valuable. I have only two major criticisms of the manuscript which relates to the section on miRNAs:

1) Crucially, the authors conclusions are based in part on analysis of a Pash1 allele, which they have engineered on the basis of the well known ts allele in C. elegans. However it does not seem clear to me how they have validated that this allele has the same effect on miRNA processing as in C. elegans. Moreover, the exact conditions they used for the permissive and non-permissive temperatures are not described in the results section and this should be added. Some validation that the allele behaves as the C. elegans allele, by high-throughput sequencing of small non-coding RNAs after shifting to the non-permissive temperature would be very useful here- it would also give insight into which miRNAs are possible candidates as, at least in C. elegans, not all miRNAs are responsive to pash-1 depletion. Small non-coding RNA sequencing is fairly straightforward now and can be outsourced to many companies without incurring huge costs. Combining this with mRNAseq analysis might further identify changes in gene expression resulting from miRNA depletion that could be linked to the neuronal phenotype (for example if one of the affected genes were cog-1 this would be consistent with the authors’ hypothesis that this is regulated by miRNAs).

2) Similarly, in the second part of this section, the conclusion of which miRNAs are involved is incomplete because mutating the putative target regions in the 3’UTR could have other effects on the transcript besides interfering with miRNA regulation. The correct way to do these experiments is to alter the target sequence by adjusting one nucleotide within the seed target region, and then if there is a phenotype, attempting to supress the phenotype by making the complementary mutation in the hypothesised miRNA’s seed sequence. This should then bring back the targeting and restore the phenotype.

There were a few minor errors in the text that I noticed, possibly some others, so it might be worth reviewing this carefully before any resubmission. One in particular is the following sentence:

“we found 1-many homologs of gcy-7 (ASEL-type) and gcy-22 156 (ASER-type) are present in the P. pacificus genome”

Do they mean ‘many homologues’ or ‘1-to-many’ homologues?

Reviewer #2: In this article, Castro et al. analyzed the evolution of a gene regulatory network regulating neuron type differentiation in nematodes. Nematodes are good model organisms to study neuronal cell type evolution as they offer a single neuron resolution. More precisely, they compared the network regulating the specification of the left and right ASE neurons between Pristionchus pacificus and C. elegans. In C. elegans, the pair of ASE neurons displays molecular and functional left-right asymmetries and the gene regulatory network that regulates this asymmetry has been well characterized. In P. pacificus, the authors have previously established that ASE left and right are functionally different, however how this asymmetry is regulated remained to be established. In this article, using HCR-FISH, a genetic screen, CRISPR genome engineering and behavioral tests, the authors characterized the gene network regulating the differentiation of the ASE left and right neurons in P. pacificus. They observed both common features as well as differences with the C. elegans network. Overall, the results are interesting and convincing. However, the presentation of the results is sometimes a bit confusing and the manuscript could be improved before publication.

Major point:

The authors examine the role of cog-1 3’UTR using CRIPR mutations and looking at the effect on terminal marker expression (gcy-7.2, gcy-22.3, gcy-8.1) by HCR-FISH. This is a great approach to characterize the phenotypic consequences. However, the interpretation, in terms of molecular mechanism, is complicated by the fact that they did not analyze the direct consequences of these cog-1 3’UTR mutations on COG-1 protein production in cis. This could be done, for example, by using their Ppa-che-1p::gfp:cog-1 3’ UTR transgene strategy. They have shown that this construct drives left-right asymmetric GFP expression in ASE. They could introduce their cog-1 3’UTR mutations in this construct and analyze the consequences on GFP expression.

Minor points:

- In fig. 1, Ppa-gcy-7.1, Ppa-gcy-7.2 and Ppa-gcy-7.3 do not cluster with Cel-gcy-7. Could the authors explain why they consider these P. pacificus genes as orthologs of C. elegans gcy-7?

- p9 lines 233-235: “The degree of reduction of gcy-8 expression in che-1 mutant animals is likely an underestimation, since only animals with at least one AFD neuron staining were scored but those without any AFD expression were not included in the overall count.” Could the authors explain why those without any AFD expression were not included in the overall count.

- p10 lines 259-269: paragraph starting with “We next ask if the other gcy-7 and gcy-22 paralogs …”. I find this paragraph a bit hard to follow, especially because the data are not shown. Showing data or adding more explanations could help.

- p14 lines 376-379: “Given the wildtype-like expression of gcy-8.1 and finger morphology in ttx-1 mutants, the primary role of ttx-1 in P. pacificus amphid neurons appears to be the repression of che-1-dependent ASE differentiation in the AFD rather than the promotion of AFD-specific fates.” This seems to me an overstatement as the ttx-1 mutant used is a hypomorph and not a null.

- p15 lines 410-411 : “We found that Ppa-pash-1(csu227) mutants express two gcy-22.3p::gfp (ASER) neurons (Fig. 10).” Fig. 10 does not show gcy-22.3p::gfp expression but HCR-FISH.

- p15 line 422: “Since in C. elegans lsy-6 miRNA repression of cog-1 is necessary for the ASER fate in the ASEL neuron” should be “Since in C. elegans lsy-6 miRNA repression of cog-1 is necessary for the ASER fate repression in the ASEL neuron”.

- p17 line 476: “the repression of ASER fate in the ASEL through binding site A1”. I do not see the evidence that this is via A1.

- Legend of fig. 3: n numbers are missing. The authors could also refer the readers to table 1.

- Fig. 5: for C and D pictures the authors should also show the gcy-7.2 and gcy-22.3 signal without the gcy-8.1 signal (as they did with A and B).

- Fig. 8C: I guess “DL” is “Dauer larvae”, this should be clarified. If this is the case then the y axis label “Dauer Formation (%)” is not correct and should be “Percentage of animals”.

- Table 3: Wildtype embryos should be added for comparison.

- SI Fig. 4: It seems that pictures A and B have been inverted.

Reviewer #3: In this manuscript, the authors dissect how asymmetry in neuronal fates is established in the satellite species P. pacificus, which allow single cell homology comparison with C. elegans, and easy gene orthology assignments. They identify how miRNAs regulate lateralised expression of cog-1, and from there lateralised expression of gcy-22 and -8 family genes.

Overall, the experiments are rational, and properly presented, although I would appreciate more details on a few points. I feel however that the main weakness of the manuscript is in the presentation and interpretation of the results.

For example, the big point here is a comparison of gene regulation between C. elegans and P. pacificus; the regulatory model for P. pacificus is nicely laid out in figure 12. I would like to see the model fo C. elegans presented side by side, to allow readers to easily compare the two.

Another main point in the title is that this asymmetric expression is convergent between C. elegans and P. pacificus. However, the fact that the regulation is different does not imply that this asymmetry evolved independently - nowhere do the authors present evidence that the ancestral species had symmetrical fate assignment for ASEL/R. A more likely possibility is that this asymmetry is ancestral (since the same nervous system seems to be highly conserved among rhabditina, and this asymmetry is but the underlying mechanisms drifted over evolutionary timescales. This difference is not trivial; mechanistically, the latter case (also termed developmental systems drift) is more interesting, since it implies regulatory changes that maintain functionality throughout, for example through redundancy. The exaptation of ASE fate specification mechanisms for AFD fate, on the other hand, seems to fundamentally differ from C. elegans.

Right now, the discussion does not really put the results in a broader context; I would appreciate a discussion on this most salient point – i.e., developmental systems drift, here illustrated by a careful dissection of gene regulation in P. pacificus.

Incidentally, it might be worth if the authors insist on the evolutionary distance between elegans and pacificus - which is larger than that with C. briggsae, on which most mechanistic comparisons with C. elegans rely.

A few more specific points:

the authors identified putative homologs via blastx search in wormbase. While this is perfectly fine as a first approach, this should be repeated with a dedicated sofware package such as Orthofinder – incidentally, if run with the genomes of other Pristionchus species, this could give information on how recent the duplications observed for gcy-7 and -22 are. Increasing the number of species can also improve the accuracy of gene trees.

In section “Multiple miRNAs are implicated in regulation of ASE and AFD fates”, the authors introduced a putative TS mutation in Ppa-pash-1. I would expect this mutation to be highly pleiotropic, disrupting many regulatory pathways. I am therefore wondering whether proper cell identity can be ascertained. How healthy are these mutants? How would the authors expect them to respond to the chemotaxis assay used on die-1 mutants?

Later on in the same section, the authors identify multiple miRNAs binding in the 3’ UTR of cog-1: mir-2251b, mir-81, mir-8353, mir-8345, mir-8364f, and identify 3 cis regulatory regions working co-operatively to drive proper expression of cog-1. In contrast, in C. elegans, only lsy-6 is described as regulating cog-1. However, there could be minor contribution from other miRNAs; have the authors checked whether there is any match for those found to be regulating cog-1 in Ppa?

Minor points:

Line 402: “since it is unclear if a functional lsy-6 mRNA homolog exists in Pristionchus…” What exactly does that mean? No clear homolog? What are the closest matches? Do you know if lsy-6 is conserved beyond the Caenorhabditis genus?

Line 289-290: it appears that gcy-22.3 and -22.5 are quite distant based on the tree in fig 1 – it is not at all unexpected that their regulation changed.

Line 813-814: Have the authors compared microsynteny between C. elegans and the two putative gcy-5 paralogs? It might be more accurate than chromosome assignment.

Figure 7: violin plots should be replaced by scatterplots, which are more representative of the actual distribution and easier to read.

Overall, the manuscript would benefit from a read-through to catch stylistic (eg. repetition of that in lines 64-65) or grammatical (eg. missing “to” at line 238) errors.

Reviewer #4: This study is important because it provides insight into the role of terminal selectors and microRNAs in the development and evolution of neuronal asymmetry. Contrary to what was assumed in the field, the paper establishes that ASE neurons of the nematode P. pacificus display asymmetrical effector gene expression. The authors dissect a miRNA-associated gene regulatory network important for ASEL/ASER and AFD identity, which is suggested to have convergently evolved relative to C. elegans. Major strengths of this work include (a) use of HCR for reliable characterization of endogenous gene expression, (b) use of both unbiased (genetic screen) and biased approaches to identify key factors acting within the novel regulatory network, (c) the use of alternative approaches to determine the importance of miRNA regulation within the evolving network despite challenges associated with miRNA homology assignment, (d) mutagenesis of endogenous miRNA binding sites with CRISPR/Cas9, and (e) the connection of misregulation of neuronal fate to behavior. The paper could be further strengthened by addressing the following issues.

Major issues:

1. The study suggests that the primary role of ttx-1 is to repress che-1-dependent ASE marker expression in the AFDs, and that ttx-1 and che-1 may interact genetically due to co-localized expression in the AFDs. This argument could be strengthened through experiments testing whether de-repression of ASE markers in a ttx-1 mutant is due to che-1 gene activity, such as through the generation of a ttx-1; che-1 double mutant.

2. The authors discuss how the mutations in putative miRNA binding sites in the cog-1 3’UTR may influence ASE asymmetry and AFD fate by changes in cog-1 expression. To obtain further mechanistic insights, experiments assessing how the 3’ UTR mutations influence reporter expression are necessary. They could help discern the regulatory function of these sites and thereby clarify how they mediate their respective phenotypes, an important point made by the authors in Discussion.

3. It is best practice to show all data mentioned in the text, rather than writing (not shown), even when the data is negative.

4. The results are somewhat disorganized, making the logic hard to follow. This could be improved by addressing the following:

- The title of each section is not always reflective of all of the results within that section. For example, the title of the second results section, “gcy-8 paralogs mark AFD neurons”, does not encapsulate the findings within that section that expression of these genes is dependent on CHE-1, or that the ASE rGC subfamily expression is also dependent on CHE-1. Similarly, “Regulators of ASE left-right asymmetry” does not reflect the behavioral findings in this section.

- It would help with readability to organize figures in the order in which they are called in the paper, both within the figure and between figures. For example, Fig 2 panels D-H are referenced in the text before Fig 2C-F. Similarly, Fig 4 A-C are discussed before Fig 3D, 3H, and 3L. Additionally, it is necessary to call out relevant panels rather than entire figures. For example, Fig 11 is called as a whole 3 times, despite the statements referencing the figure only discussing specific panels.

- The Introduction, Results, and Discussion can be shorter, more succinct. Throughout the text, the language can be improved by being more precise and less redundant. Phrases like “invisible hands of miRNA regulation” should be eliminated. The phrase “familiar guild of TFs interacting in unfamiliar ways” does not provide any substantial information.

Minor issues:

It would be helpful to color code the first figure to match the color code used throughout the rest of the paper (AFD in purple, ASEL in red, ASER in green).

In the legend of Figure 3, the n= values are missing.

The summary schematic of Figure 7 ignores potential contributions of the AFD neurons to behavior.

Figure 12: It would be helpful to the reader to include a schematic of the C. elegans ASE regulatory network (bistable feedback loop) for comparison.

**Have all data underlying the figures and results presented in the manuscript been provided?**

Reviewer #1: None

Reviewer #2: None

Reviewer #3: Yes

Reviewer #4: Yes

PLOS authors have the option to publish the peer review history of their article (what does this mean? ). If published, this will include your full peer review and any attached files.

**Do you want your identity to be public for this peer review?** For information about this choice, including consent withdrawal, please see our Privacy Policy .

Reviewer #1: No

Reviewer #2: No

Reviewer #3: No

Reviewer #4: **Yes:** Paschalis Kratsios

**Figure resubmission:**
---

## [Decision Letter · Decision Letter 1]

7 Nov 2025

PGENETICS-D-25-00733R1

The rewiring of a terminal selector regulatory cascade generates convergent neuronal laterality

PLOS Genetics

Dear Dr.Ray L. Hong,

Thank you for submitting your manuscript to PLOS Genetics. After careful consideration, we feel that it has merit but does not fully meet PLOS Genetics's publication criteria as it currently stands. Therefore, we invite you to submit a revised version of the manuscript that addresses the points raised during the review process.

Please submit your revised manuscript within by Dec 07 2025 11:59PM. If you will need more time than this to complete your revisions, please reply to this message or contact the journal office at plosgenetics@plos.org. Please include the following items when submitting your revised manuscript:

We look forward to receiving your revised manuscript.

Kind regards,

Nathalie Pujol

Academic Editor

PLOS Genetics

Monica Colaiácovo

Section Editor

PLOS Genetics

Aimée Dudley

Editor-in-Chief

PLOS Genetics

Anne Goriely

Editor-in-Chief

PLOS Genetics

**Additional Editor Comments:**

The authors need to address the issue raised by Reviewer 1 regarding the pash-1 allele

**Reviewers' comments:**

Reviewer's Responses to Questions

**Comments to the Authors:**

Reviewer #1: I thank the authors for their response to the first line of questions. I'm mostly happy but I'm afraid that I don't accept the authors' response to my point about validating that the pash-1 ts allele leads to loss of miRNAs genome-wide. Without confirming this the experiment cannot be interpreted, so I think it is very much inside the scope of the manuscript. If it is not possible to do high-throughput sequencing of small non-coding RNAs (although this is really no longer prohibitively expensive, as I pointed out in my previous review) the authors could consider northern blot analysis of several miRNAs or qPCR analysis of a panel of miRNAs.

Reviewer #2: The authors have addressed all the points that I raised in a satisfactory manner.

Reviewer #3: Overall, the authors improved substantially the manuscript with their revisions. It is a bit long, but not excessively so. Most of the recommendations I had were properly adressed, save for a couple:

the Ppa-pash-1 phenotype is, as expected, highly pleiotropic. I feel that the authors should mention this in the manuscript, and maybe explain why they are confident about the fact that the cells are there, but misspecified (it is not immediately obvious based on the figures). The authors may also mention that the worms are too sick to perform assays on.

A quick mention of the fact that most functional comparisons were between C. elegans and C. briggsae, and how P. pacificus is much more divergent - put an estimate of divergence time, so that non-worm people can have an idea of the significance of functional conservation.

Finally, I would encourage the authors to read through very carefully to catch minute errors - eg. misformated reference, as on page 22 with Ahmed et al. without a publication year, or a few misplaced spaces.

Reviewer #4: The authors have addressed all my concerns. The revised manuscript is very much improved.

**Have all data underlying the figures and results presented in the manuscript been provided?**

Reviewer #1: Yes

Reviewer #2: None

Reviewer #3: Yes

Reviewer #4: Yes

PLOS authors have the option to publish the peer review history of their article (what does this mean? ). If published, this will include your full peer review and any attached files.

**Do you want your identity to be public for this peer review?** For information about this choice, including consent withdrawal, please see our Privacy Policy .

Reviewer #1: No

Reviewer #2: No

Reviewer #3: No

Reviewer #4: **Yes:** Paschalis Kratsios

**Figure resubmission:**
---

## [Decision Letter · Decision Letter 2]

20 Jan 2026

PGENETICS-D-25-00733R2

The rewiring of a terminal selector regulatory cascade generates convergent neuronal laterality

PLOS Genetics

Dear Dr. Ray L. Hong,

Thank you for submitting your manuscript to *PLOS Genetics* . We are pleased to inform you that your study has been provisionally accepted for publication, pending the deposit of the raw sequencing data in a public repository such as the Sequence Read Archive (SRA) or an equivalent platform.

Once you have completed this step and provided us with the accession number, and included in the MS, please submit your revised manuscript within by Feb 19 2026 11:59PM. If you will need more time than this to complete your revisions, please reply to this message or contact the journal office at plosgenetics@plos.org. Please include the following items when submitting your revised manuscript:

We look forward to receiving your revised manuscript.

Kind regards,

Nathalie Pujol

Academic Editor

PLOS Genetics

Monica Colaiácovo

Section Editor

PLOS Genetics

Aimée Dudley

Editor-in-Chief

PLOS Genetics

Anne Goriely

Editor-in-Chief

PLOS Genetics

**Additional Editor Comments:**

Please deposit the raw small RNA sequencing data in the SRA or a comparable public repository to meet data availability requirements before final acceptance.

**Journal Requirements:**

**Reviewers' comments:**

Reviewer's Responses to Questions

**Comments to the Authors:**

Reviewer #1: Thank you for accommodating my final request. The small RNA sequencing confirms the effect of the pash-1 allele. The only remaining thing to do is to deposit the raw data for small RNA sequencing as indicated in the section below on raw data and reference to these made in the data availability section. I'm happy with publication after this has been done.

Reviewer #3: All concerns have been adressed; this manuscript can be published in its present form.

**Have all data underlying the figures and results presented in the manuscript been provided?**

Reviewer #1: **No:** The raw data for the small RNA sequencing has not been deposited. It needs to be made available via SRA or similar.

Reviewer #3: Yes

PLOS authors have the option to publish the peer review history of their article (what does this mean? ). If published, this will include your full peer review and any attached files.

**Do you want your identity to be public for this peer review?** For information about this choice, including consent withdrawal, please see our Privacy Policy .

Reviewer #1: No

Reviewer #3: No

**Figure resubmission:**
---

## [Editor Report · Decision Letter 3]

1 Feb 2026

Dear Dr Hong,

We are pleased to inform you that your manuscript entitled "The rewiring of a terminal selector regulatory cascade generates convergent neuronal laterality" has been editorially accepted for publication in PLOS Genetics. Congratulations!

Yours sincerely,

Nathalie Pujol

Academic Editor

PLOS Genetics

Monica Colaiácovo

Section Editor

PLOS Genetics

Aimée Dudley

Editor-in-Chief

PLOS Genetics

Anne Goriely

Editor-in-Chief

PLOS Genetics

BlueSky: @plos.bsky.social

Comments from the reviewers (if applicable):

**Data Deposition**

http://datadryad.org/submit?journalID=pgenetics&manu=PGENETICS-D-25-00733R3

**Press Queries**

---

## [Editor Report · Acceptance letter]

PGENETICS-D-25-00733R3

The rewiring of a terminal selector regulatory cascade generates convergent neuronal laterality

Dear Dr Hong,

We are pleased to inform you that your manuscript entitled "The rewiring of a terminal selector regulatory cascade generates convergent neuronal laterality" has been formally accepted for publication in PLOS Genetics! Your manuscript is now with our production department and you will be notified of the publication date in due course.

With kind regards,

Anita Estes

PLOS Genetics

On behalf of:
